# Roadmap of phase transitions in hafnia-based superlattice films

Wan-Rong Geng [1,2,7], Bo-Rui Wang[1,3,7], Yin-Lian Zhu [1,2,4], Si-Rui Zhang[3], Min Liao[3] & Xiu-Liang Ma [1,2,5,6] ✉

Hafnia-based ferroelectrics hold significant promise for next-generation non-volatile memory. However, their functional properties are critically limited by uncontrollable phase transitions due to the poorly understood atomistic mechanisms driving specific transformations. Here, using single-crystalline HfO$_2$-based superlattice films as the prototype system, we propose an asynchronous sublattice distortion mechanism underlying the complex phase transitions in HfO$_2$-based materials. Aberration-corrected transmission electron microscopy reveals that sublattice preferential distortion behaviors trigger various phase transitions among orthorhombic, tetragonal and monoclinic phases, processes governed by the direction of orthorhombic phase. Critically, the complex lattice distortion pathways underlying the orthorhombic-to-monoclinic transition are elucidated, revealing their fundamental dependence on the monoclinic projection direction. Furthermore, polar-antipolar transition within the orthorhombic phase requires only oxygen sub-lattice dipole-order reversal, enabling polarization flipping. This work systematically clarifies the core mechanisms of structural phase transitions in HfO$_2$-based films, resolving previous controversies and providing a guidance for designing high-performance HfO$_2$-based electronic devices.

Since the discovery of ferroelectricity in hafnia-based binary oxide (HfO$_2$-based) systems, this material has attracted considerable interest over the past decade due to its compatibility with CMOS processing and robust ferroelectricity at the nanoscale[1–5]. However, unlike perovskite-typed ferroelectric thin films, in which a simple and stable ferroelectric phase structure, and rich ferroelectric topological domain configurations have been identified, benefiting from mature single-crystal epitaxial growth techniques[6–9], HfO$_2$-based ferroelectrics exhibit multiple polymorphic phases[10,11], including orthorhombic (*O*-phase), monoclinic (*M*-phase), and tetragonal (*T*-phase)[12,13]. Especially, its ferroelectric phase is metastable *O*-phase under ambient conditions and readily transforms into thermodynamically stable phases such as the *M*-phase or *T*-phase[14,15]. This instability typically results in

polycrystalline, multiphase films, severely constraining the stability and memory performance of HfO$_2$-based ferroelectric devices. Consequently, achieving strong stabilization of the ferroelectric phase and elucidating its complex phase transformation mechanisms have emerged as critical research priorities.

Previous studies have successfully stabilized the metastable ferroelectric phase and modulated ferroelectric properties through diverse approaches, including elemental doping[16,17], ultrathin layer design[2,18], strain engineering[19], epitaxial orientation control[20], and surface/interface engineering[21–23]. Furthermore, diverse structural phase transitions have been observed in different systems of HfO$_2$-based ferroelectrics via external field stimulation, yet a consistent understanding remains elusive. For instance, electron-beam

[1]Bay Area Center for Electron Microscopy, Songshan Lake Materials Laboratory, Dongguan, China. [2]Dongguan Institute of Materials Science and Technology, Chinese Academy of Sciences, Dongguan, China. [3]School of Advanced Materials and Nanotechnology, Xidian University, Xi'an, China. [4]School of Materials Science and Engineering, Hunan University of Science and Technology, Xiangtan, China. [5]Institute of Physics, Chinese Academy of Sciences, Beijing, China. [6]Quantum Science Center of Guangdong-HongKong-Macau Greater Bay Area, Guangdong Shenzhen, China. [7]These authors contributed equally: Wan-Rong Geng, Bo-Rui Wang. ✉e-mail: xlma@iphy.ac.cn

irradiation has been shown to drive phase transformations between the *M*-phase and *O*-phase, and between the *T*-phase and *O*-phase in freestanding $ZrO_2$ thin films[14]. Structural linkages between ferroelastic switching and the *O*-to-*M* transition have been elucidated in $ZrO_2$ nanocrystals[15]. However, existing researches on the *O*-*M* phase transition has predominantly focused on the lattice shear behavior of the cation sublattice and the corresponding strain evolution, while overlooking the role of the anions (oxygen ions), thus leaving the underlying structural mechanism incompletely understood. Furthermore, the specific sequence of cation- and anion-sublattice distortions during the *T*-*O* phase transition remains a subject of debate, as experimental findings across different $HfO_2$-based systems are inconsistent[14,24]. Beyond these fundamental transitions, the structural origins of wake-up and fatigue under electric fields, though often attributed to phase transitions, remain controversial. For instance, structural studies on $Hf_{0.5}Zr_{0.5}O_2$ capacitors attribute these phenomena to reversible transitions between ferroelectric (*FE*) and antiferroelectric (*AFE*) *O*-phases[25]. In contrast, investigations on $HfO_2$ capacitors link wake-up to changes in the phase structure (involving *M*-, *O*-, and *T*-phases) in the bulk and at interfaces, while ascribing fatigue to defect generation and domain-wall pinning[26].

In particular, these findings are dispersed across various forms of fluorite-structured materials, including polycrystalline films, nanocrystals, and bulk materials, and across differently doped $HfO_2$-based systems. Discrepancies in structure, doping concentrations, complex interfacial architectures, and high specific surface areas among these systems can significantly influence differences in phase transition behaviors. This has led to divergent interpretations of the underlying phase transformation mechanisms[14,15,24,25,27,28], thus hindering a unified understanding and impeding the development of rational strategies to stabilize the metastable ferroelectric phase and optimizing device performance. Therefore, systematically uncovering the complex phase transition behaviors and their structural mechanisms in $HfO_2$-based thin films is essential to provide comprehensive structural insights for material design.

In this study, we eliminated the influence of complex interfaces inherent in polycrystalline films by constructing a simplified model system: $HfO_2$-based superlattice single-crystal films. Based on the in-situ electron-beam irradiation experiment using an aberration-corrected transmission electron microscope, we revealed the step-by-step sublattice distortion mechanism of various phase transitions, including *T*-to-*O* phase transitions, *M*-to-*O* phase transitions with varying orientations, and structural transformations between *FE* and *AFE* states within the *O*-phase itself. Through detailed structural analysis at the unit-cell scale, distinct sublattice distortion modes associated with each transition pathway are summarized. Especially, the transformation of the *O*-phase into *M*-phases of different orientations involves distinct activation conditions and ion displacement patterns. The initial crystallographic orientation of the *O*-phase is found to dictate its selective transformation pathway towards either the *T*-phase or *M*-phase. This work provides a comprehensive, systematic structural framework for understanding phase-transition behavior in $HfO_2$-based films, facilitating the design of high-performance hafnia-based ferroelectric devices.

## Results

Given that $Y_2O_3$:$ZrO_2$ (YSZ) substrates share the fluorite structure with $HfO_2$- and $ZrO_2$-based materials, the epitaxial growth of $HfO_2$-based

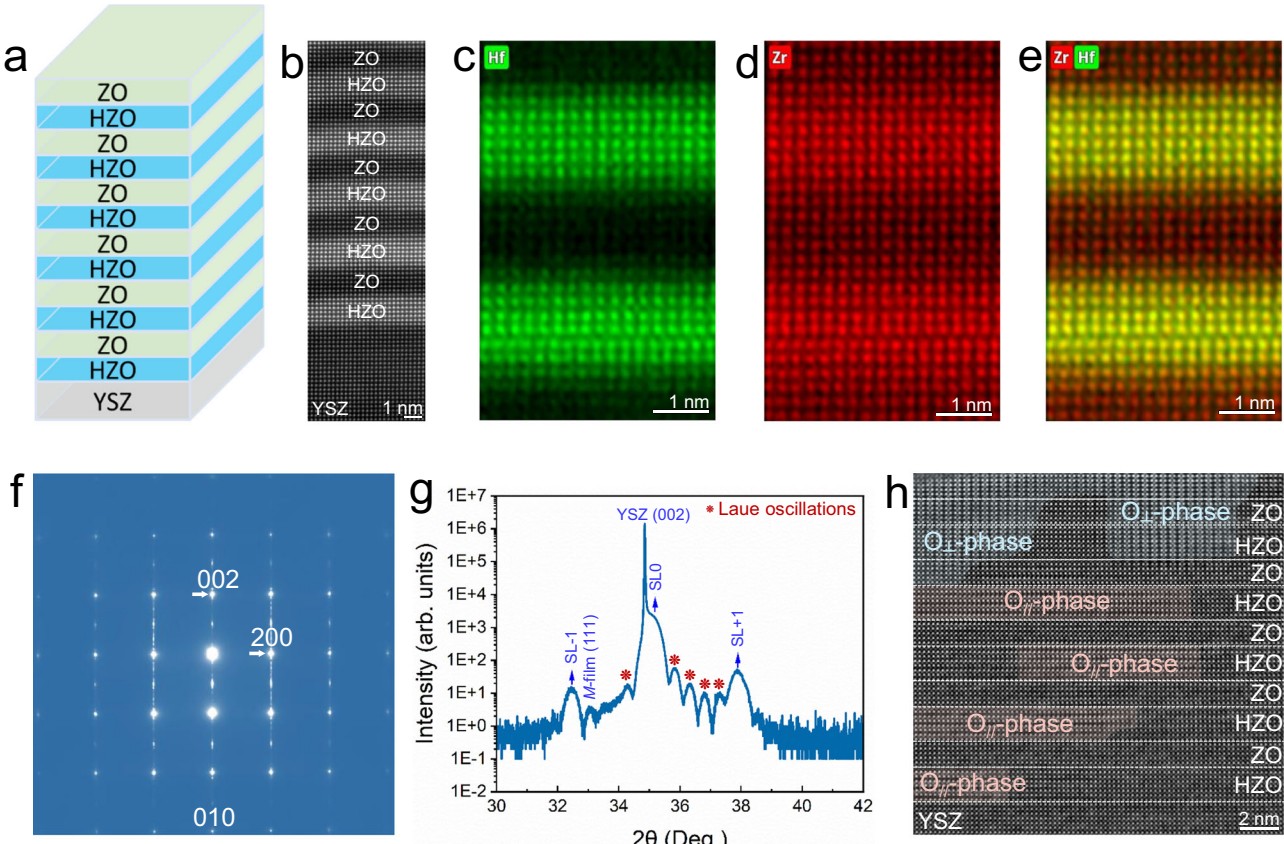

**Fig. 1 | (HZO-ZO)₆/YSZ (001) superlattice film. a** Schematic of the superlattice film. **b** HAADF-STEM image displaying the high-quality growth of superlattice films. **c-e,** Atomic-resolved EDS elemental maps of Hf (**c**), Zr (**d**) and Hf and Zr combined distribution (**e**). **f** The SAED pattern corresponding to the region including both the superlattice film and YSZ substrate. **g** High-resolution XRD *θ-2θ* scan of the film, suggesting the superlattice periodicity and high-quality film growth. **h** Atomic-resolved iDPC-STEM image, showing two kinds of *O*-phase: one aligned horizontally within the HZO layer (*O∥*-phase, pink mask), and the other traversing vertically through both the HZO and ZO layers (*O⊥*-phase, cyan mask).

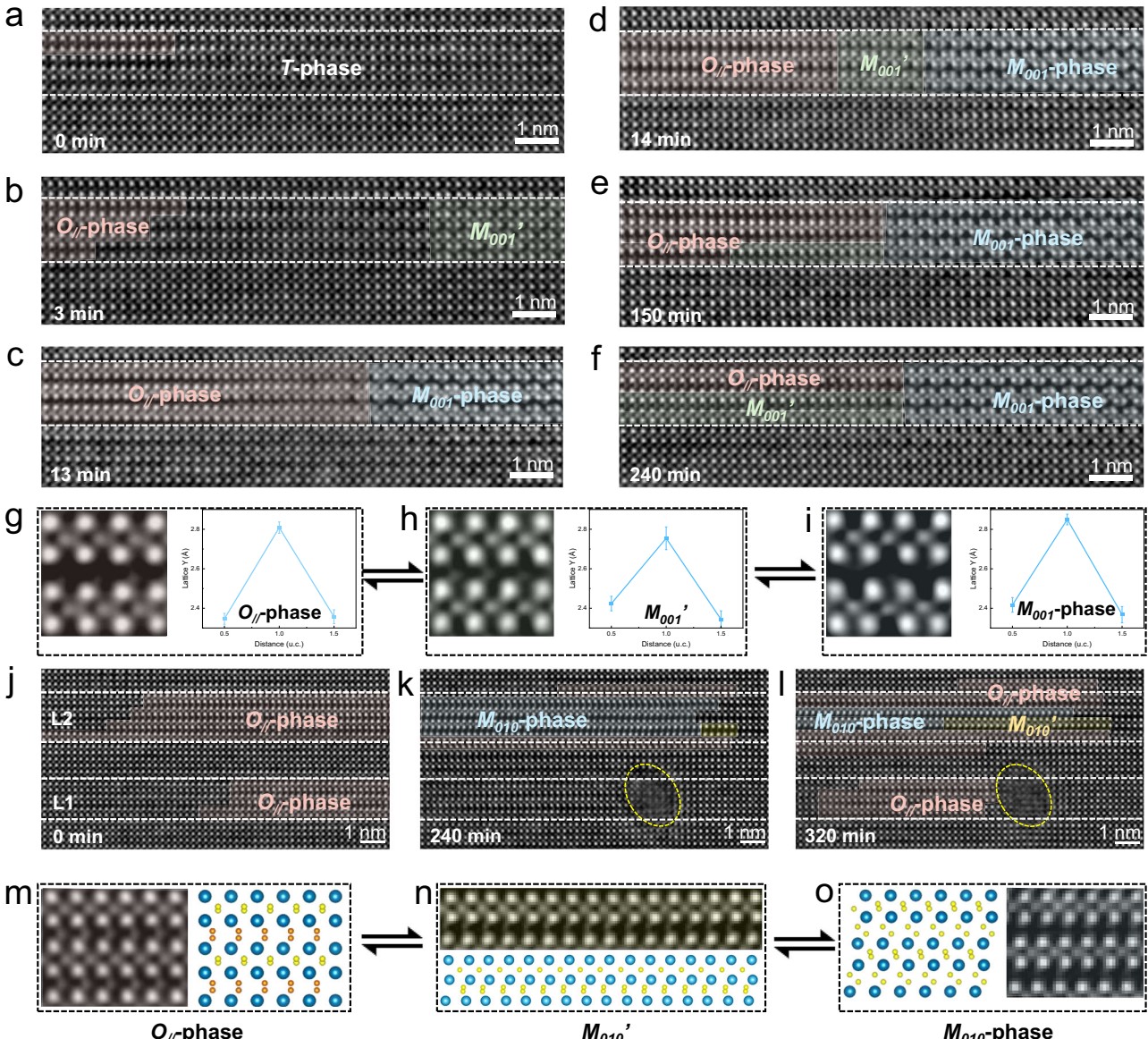

**Fig. 2 | Phase transition among *T*-phase, *O*-phase and *M*-phase. a–f** Atomic-scale iDPC-STEM images after different irradiation durations: 0 min (**a**); 3 min (**b**); 13 min (**c**); 14 min (**d**); 150 min (**e**); 240 min (**f**), with the *O*-phase, intermediate state ($M_{001}'$) and $M_{001}$-phase highlighted by pink, green and cyan masks, respectively. **g–i** Enlarged iDPC-STEM images of the *O*-phase, $M_{001}'$ and $M_{001}$-phase (left panel) and corresponding Lattice Y maps (right panel), respectively. **j–l** Atomic-scale iDPC-STEM images after different irradiation durations: 0 min (**j**); 240 min (**k**); 320 min (**l**), with the *O*-phase, intermediate state ($M_{010}'$) and $M_{010}$-phase highlighted by pink, yellow and cyan masks, respectively. **m** Enlarged iDPC-STEM image of *O*-phase and its unit cell schematic. **n** Enlarged iDPC-STEM image of $M_{010}'$ and its unit cell schematic. **o** Enlarged iDPC-STEM image of $M_{010}$-phase and its unit cell schematic.

thin films on YSZ substrates enables superior crystalline quality. Furthermore, superlattice thin-film systems, as typical size-confined systems, are constructed by periodically alternating the growth of different materials, thereby displaying strong interfacial and interlayer interactions. They are expected to overcome the physical limitations of bulk materials, for example, stabilizing the desired polar structures and inducing novel physical properties in $HfO_2$-based materials[29,30]. Thus, using pulsed laser deposition (PLD), we fabricated a series of high-quality ($Hf_{0.5}Zr_{0.5}O_2$-$ZrO_2$)$_6$ (($HZO$-$ZO$)$_6$) epitaxial superlattice thin films on [001]-oriented YSZ substrates with different thickness of one (HZO-ZO) growth period, including the 5 unit cells (labeled as HZO-ZO-5), 6 unit cells (labeled as HZO-ZO-6) and 11 unit cells (labeled as HZO-ZO-11). In Fig. 1a, b, the (HZO-ZO-6)$_6$ superlattice film is displayed. Atomic-scale EDS-STEM mapping revealed sharp interlayer interfaces (Fig. 1c–e), attesting to the high-quality growth. The in-

depth analysis of the cation/anion stoichiometry for the superlattice film is further shown in Supplementary Fig. 1 and Supplementary Tables 1 and 2. The results indicate a slight interdiffusion of Hf into the neighboring $ZrO_2$ layer, along with a minor oxygen deficiency in the HZO layer. This slight off-stoichiometry and elemental interdiffusion are likely due to the high-temperature growth process used for the superlattice film. Selected-area electron diffraction (SAED) further validated the single-crystallinity nature of the superlattice (Fig. 1f). Based on the based diffraction spots in the SAED pattern of cubic YSZ (001) substrate (Supplementary Fig. 2), the coexistence of *M*-phase, *O*-phase and *T*-phase is confirmed in the hafnia-based superlattice film (Supplementary Fig. 3). The schematic diagrams for the multiple phases in hafnia-based materials are presented in Supplementary Fig. 4. High-resolution X-ray diffraction (XRD) scan (Fig. 1g) and wide-angle XRD scan (Supplementary Fig. 5) reveal the single-crystalline

nature of the superlattice and the well-defined superlattice periodicity and high-quality film growth. Besides, the X-ray reflectivity in Supplementary Fig. 6 shows clear Kiessig fringes, further confirming the high-quality growth of the superlattice film[30]. From the period of the Kiessig fringes, the total film thickness is estimated to be approximately 20.5 nm. The distinct superlattice Bragg peak observed at $2\theta = 2.67°$ corresponds to a superlattice period of about 3.3 nm, indicating a well-defined periodic chemical modulation. Moreover, atomic-resolution integrated differential phase contrast scanning transmission electron microscopy (iDPC-STEM) imaging demonstrated a dislocation-free HZO/YSZ heterointerface and established a coherent epitaxial relationship between the superlattice and substrate (Fig. 1h). Furthermore, the iDPC-STEM image reveals two dominant phase structures in the film: a $T$-phase and an $O$-phase. The presence of the polar phase is further confirmed by the second harmonic generation results in Supplementary Fig. 7. The $O$-phase exhibits two distinct orientations (Fig. 1h): (i) horizontally aligned domains predominantly within the HZO layer ($O_{//}$-phase, pink mask), with the adjacent ZO layer maintaining a nonpolar $T$-phase structure; (ii) vertically oriented domains traversing both HZO and ZO layers ($O_{\perp}$-phase, cyan mask). It is revealed that the phase structure, spatial distribution, and epitaxial quality of the superlattice films depend on the thickness of one (HZO-ZO) period (Supplementary Figs. 8 and 9). The particular (HZO-ZO)$_6$ film investigated here, with a period thickness of 6 unit-cells, exhibits both high growth quality and a stabilized polar $O$-phase.

HfO$_2$-based superlattice films inherently possess a rich polymorphism, encompassing $O$-, $T$-, and $M$-phases (Fig. 1). This inherent phase complexity can lead to poor ferroelectric performance with evident leakage current (Supplementary Fig. 10) and pose a risk of field-induced phase transitions causing device failure. Therefore, elucidating the detailed transition pathways and mechanisms between different phases driven by external stimuli is crucial for enhancing the stability of ferroelectric properties in these materials. Electron-beam irradiation, an established approach for triggering structural dynamics in electron microscopy, has been demonstrated to induce domain reconfiguration[31,32], phase transitions[14,15], and defect formation[33].

As the crucial $O$-phase in the hafnia-based materials, the polar and anti-polar $O$-phases display the distinct structural characteristics of one-wide-one-narrow Hf/Zr sublattice distribution (Supplementary Fig. 4) accompanied with the non-centrosymmetric displacement of oxygen atoms, with the wide sublattice being the ferroelectric layer and narrow sublattice being the spacer layer. Thus, two kinds of oxygen atoms are also denoted, including the O$_1$ in the ferroelectric layer and O$_2$ in the spacer layer, as shown in Supplementary Fig. 4. The evident structural features play a vital role in determining the phase transition between $O$-phase and other nonpolar phases in the hafnia-based materials. Complementary analysis in Fig. 2 elucidates the transition mechanisms between the $O$-phase and $M$-phase (Schematics in Supplementary Fig. 11), which is crucial for understanding and optimizing the ferroelectric switching behavior of HfO$_2$-based films, and is pivotal in addressing device reliability and performance degradation issues. The dominant structure in HZO layers in the as-grown superlattice film (Fig. 2a) presents the nonpolar cubic lattice, which may correspond to either the cubic ($C$) phase or $T$-phase. Since the theoretical energy of the $C$-phase is much higher than that of $T$-phase[34], the initial structure of the film is considered to be $T$-phase. Figure 2a–f tracks the phase evolution among $T$-, $O_{//}$-, and $M$-phases in the HZO layer under electron beam irradiation. The predominant pathway is that $T$-phase progressively transforms into $O_{//}$-phase (pink mask) with increasing irradiation time, followed by the emergence of an intermediate state $M_{OO1}'$ (green mask), and ultimately the formation of the $M$-phase projected along [001] direction ($M_{OO1}$, cyan mask). This transition pathway is consistent with previous reports of density functional theory calculations. It was revealed that the energy required for the $T$-$O$ transformation was much lower than that for a direct

transition from $T$-phase to $M$-phase under the fixed lattice parameters of HfO$_2$ ref. 35. Crucially, the transformation from $O$-phase to $M_{OO1}$-phase proceeds through an intermediary phase, with the $O$-phase first evolving into the $M_{OO1}'$ phase before converting to the final $M_{OO1}$ phase. The $M_{OO1}'$ phase differs from $M_{OO1}$ in its lattice distortion, as evidenced in Supplementary Fig. 12. Understanding the sequence of cationic (Hf/Zr) and anionic (O) sublattice distortions during this process is key to elucidating the underlying mechanism.

High-resolution iDPC-STEM images (left panels) and corresponding out-of-plane lattice constant (Lattice Y) spacing profiles (right panels) of the $O$-phase, $M_{OO1}'$-state, and $M_{OO1}$-phase are presented in Fig. 2g–i. Both $O$- (Fig. 2g) and $M_{OO1}$- (Fig. 2i) phases exhibit an asymmetric wide-narrow Hf/Zr sublattice spacing pattern, but with critical distinctions. Besides the unidirectional oxygen displacement in $O$-phase, the wide/narrow Hf/Zr sublattice distances (2.81 Å/2.35 Å) in $O$-phase are smaller than these values in $M_{OO1}$-phase (2.85 Å/2.39 Å). Notably, the $M_{OO1}'$-state (Fig. 2h) displays a hybrid character. It retains the unidirectional oxygen displacement of the $O$-phase but adopts a modified Hf/Zr sublattice spacing of 2.75 Å/2.38 Å. This indicates that the $O$-$M_{OO1}$ transition is initiated by a distortion of the Hf/Zr cation sublattice, forming the intermediate $M_{OO1}'$ state. In this state, the wide Hf/Zr sublattice distance decreases while the narrow distance increases. Given that the final $M$-phase exhibits larger Hf/Zr spacings, it is inferred that $M_{OO1}'$ undergoes a subsequent, complex lattice reconstruction to reach the final $M_{OO1}$ structure-a potentially rapid, transient step not captured here.

Beyond the $O$-$M$ phase transition pathway shown in Fig. 2g–i, it is discovered that electron-beam-induced local defects (yellow dashed ellipses, Fig. 2j–l) can facilitate a distinct, reversible $O$-$M_{O1O}$ transition ($M_{O1O}$ denoting the $M$-phase projected along [010] direction). This alternative pathway is mediated by the localized strain fields generated by these defects. The initial state (Fig. 2j) shows locally stable $O_{//}$-phase (pink mask). After 240 min irradiation (Fig. 2k), a defect forms in layer 1 (L1), concomitant with $O$-phase loss in L1 and the nucleation of a two-unit-cell $M_{O1O}$ domain (cyan mask) in layer 2 (L2). This suggests that defect-induced shear strain drives the $O$-$M_{O1O}$ phase transition in L2. Further irradiation to 320 min (Fig. 2l) triggers $O$-phase reformation near the defect in L1, while part of the $M_{O1O}$-phase in L2 transforms into an intermediate state (yellow mask), signaling an incipient $M_{O1O}$-$O$ transition. Figure 2m–o provides atomic-scale iDPC-STEM images and corresponding unit cell schematics to dissect this $M_{O1O}$-$O$ process. The intermediate state (Fig. 2n) features that Hf/Zr sublattice configuration remains characteristic of the $M_{O1O}$-phase, while oxygen displacement pattern shifts to the unidirectional mode of the $O$-phase. This demonstrates an oxygen-first mechanism for the $M_{O1O}$-$O$ transition, where the oxygen ions displace first to form the intermediate state and followed by reorganization of the cation sublattice. In contrast, the Hf/Zr-first mechanism contributes to the intermediate state and then realizes the $O$-$M_{O1O}$ phase transition. It is worthwhile to note that the spatial resolution limit of iDPC-STEM imaging precludes resolving closely spaced oxygen ions. As evident in Fig. 2m, o, while schematics (right panels) show two adjacent oxygen ions at the center of the narrow Hf/Zr sublattice, the experimental images (left panels) reveal only a single intensity maximum between four Hf/Zr columns. This limitation applies equally to $M_{O1O}'$-state imaging. These results elucidate distinct ion-displacement pathways that govern the reversible $O$-$M_{O1O}$ phase transformations in HfO$_2$-based oxides, demonstrating their critical dependence on the local strain environment.

It is shown that the phase from $O_{//}$-phase to $M_{OO1}$-phase initiates from the as-grown $T$-phase in Fig. 2a–f, forming a sequential $T$-$O_{//}$/$M_{OO1}$ pathway. In contrast, the transition between $T$-phase and $O_{\perp}$-phase exhibits markedly different behavior under electron beam irradiation. Supplementary Fig. 13 presents a series of iDPC-STEM images demonstrating the reversible phase transition between the $T$- and $O_{\perp}$-phases in a HfO$_2$-based superlattice film under electron beam

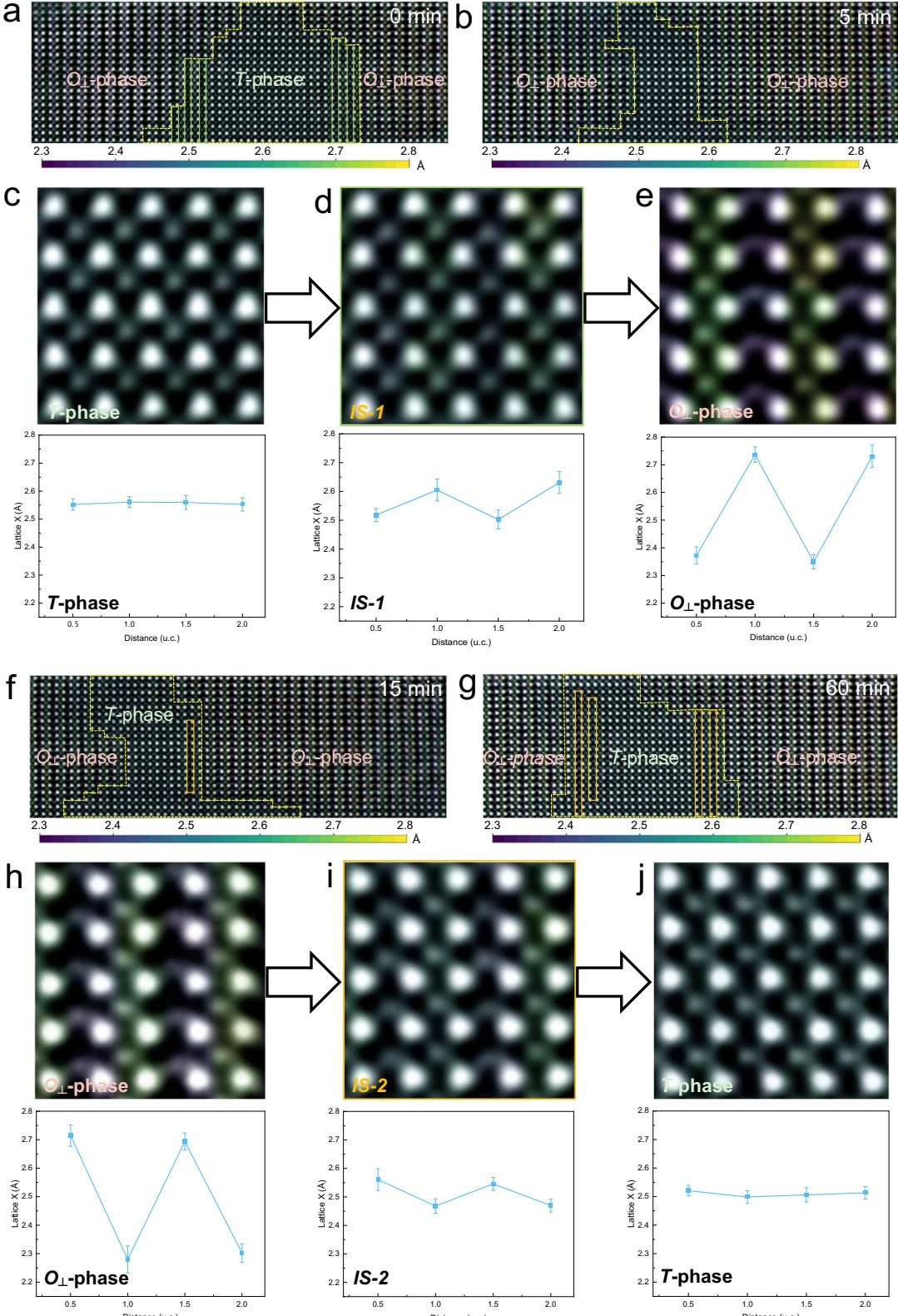

**Fig. 3 | Reversible phase transition between *T*-phase and *O*-phase. a, b** Overlay results of the in-plane lattice parameter (Lattice X) map and iDPC-STEM image during the *T-O* phase transition. Dashed yellow lines denote the phase boundaries, and green rectangles denote the intermediate state *IS-1*. **c**–**e** Enlarged iDPC-STEM images of the *T*-phase, *IS-1*, and *O*-phase (upper panel) and corresponding Lattice X maps (lower panel). **f**–**g** Overlay results of the in-plane lattice parameter map and iDPC-STEM image during the *O-T* phase transition. Dashed yellow lines denote the phase boundaries, and orange rectangles denote the intermediate state *IS-2*. **h**–**j** Enlarged iDPC-STEM images of the *O*-phase, *IS-2*, and *T*-phase (upper panel) and corresponding Lattice X maps (lower panel).

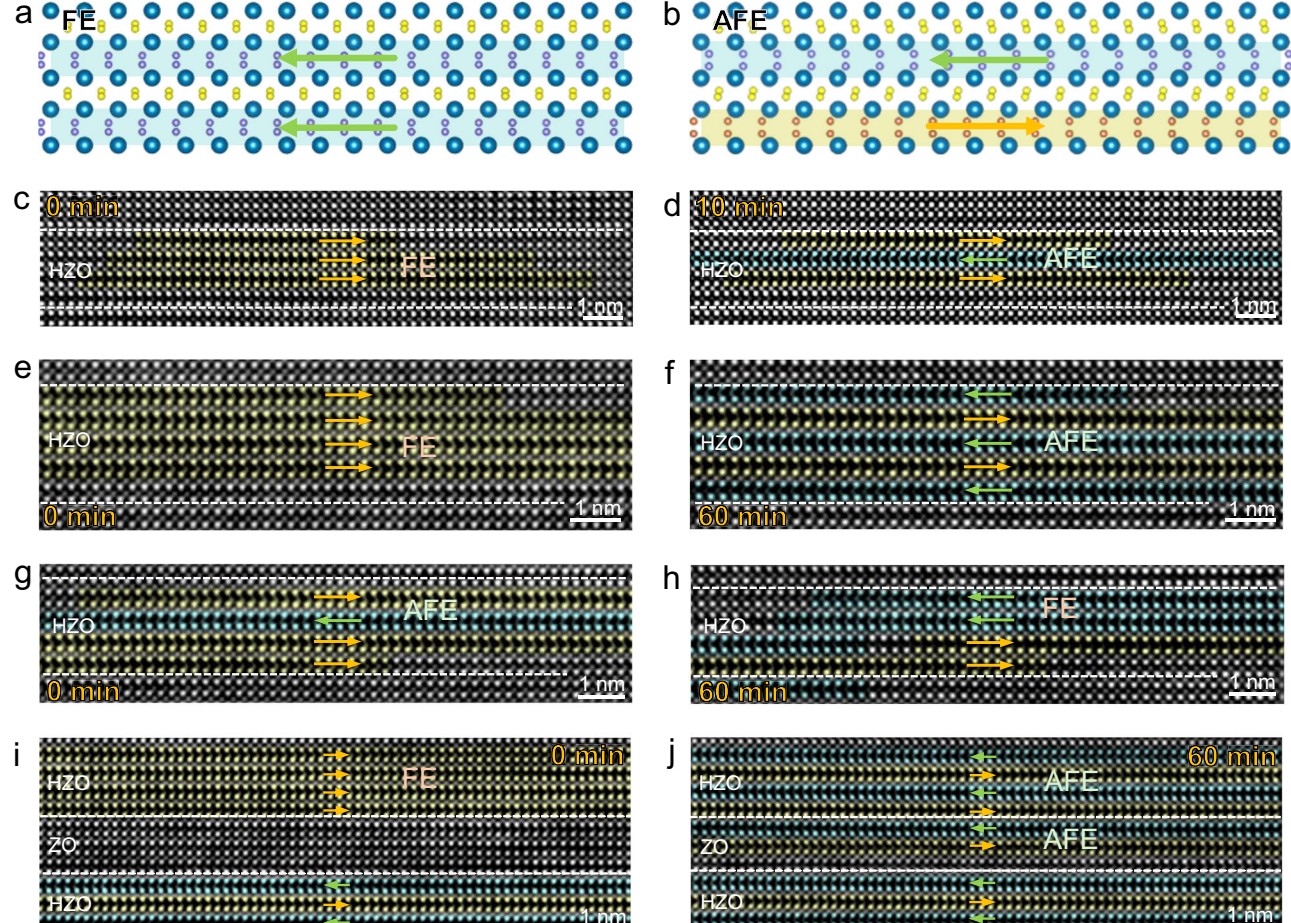

**Fig. 4 | Real-time phase transition behaviors between *FE-O* phase and *AFE-O* phase under electron beam irradiation. a** Schematic diagram of the *FE-O* phase. **b** Schematic diagram of the *AFE-O* phase. **c-d**, One-unit-cell *FE-AFE* transition after 10 min electron beam illumination. **e, f** Three-unit-cell *FE-AFE* transition after 60 min electron beam illumination. **g, h** One-unit-cell *AFE-FE* transition after 60 min electron beam illumination. **i, j** Stabilization of horizontal *AFE-O* phase in the ZO layer after 60 min electron beam illumination.

irradiation. During the initial 45 min of irradiation (Supplementary Fig. 13a–c), the proportion of the $O_\perp$-phase progressively increases while the *T*-phase concurrently decreases (cyan mask), indicating a *T*-$O_\perp$ phase transformation. Upon further irradiation (Supplementary Fig. 13d–f), the $O_\perp$-phase proportion diminishes, and the *T*-phase recovers, signifying the reverse $O_\perp$-*T* phase transition. Notably, after 360 min of irradiation (Supplementary Fig. 13f), the *T*-phase becomes predominant.

The structural distinctions between the *T*- and $O_\perp$-phases primarily reside in the distribution of Hf/Zr sublattice and the presence (or absence) of oxygen ion displacements. Consequently, determining the sequential order of these lattice distortions during the phase transitions is essential, which would provide critical guidance for designing phase transformation pathways and stabilizing the *O*-phase. Figure 3a, b illustrates the *T*-$O_\perp$ transition process, combining iDPC-STEM images with in-plane lattice constant (Lattice X) maps. A distinct intermediate state (*IS-1*) during this transformation is highlighted by the green rectangle in Fig. 3a. Correspondingly, Fig. 3f–g present the $O_\perp$-*T* phase transition, with an intermediate state (*IS-2*) marked by the orange rectangle in Fig. 3g. Figure 3c–e presents high-resolution iDPC-STEM images (top row) and corresponding Lattice X spacing profiles (bottom row) of the *T*-phase, *IS-1*, and $O_\perp$-phase, revealing their lattice characteristics and the sequence of lattice distortions during the *T*-$O_\perp$ transition. The *T*-phase (Fig. 3c) exhibits no oxygen displacement and a uniform Lattice X distribution, with the averaged Hf/Zr sublattice

distance being 2.56 Å. In contrast, the $O_\perp$-phase (Fig. 3e) displays a characteristic one-wide-one-narrow Lattice X distribution (2.36 Å/2.73 Å), accompanied by significant oxygen displacement at the wide Hf/Zr sublattice layers. The intermediate state *IS-1* (Fig. 3d) also shows a one-wide-one-narrow Lattice X pattern (2.51 Å/2.62 Å), but with markedly lower amplitude in the wide Hf/Zr sublattice and higher amplitude in the narrow Hf/Zr sublattice than that of the $O_\perp$-phase and exhibits no oxygen displacement. The results demonstrate that the *T*-$O_\perp$ transition proceeds via cation sublattice distortion preceding anion displacement: The Hf/Zr cation sublattice first distorts from *T*-phase, manifested as contraction in one sublattice and expansion in the other, forming the intermediate state *IS-1* characterized by the nascent wide-narrow Lattice X pattern without oxygen displacement. Subsequently, oxygen ions displace, culminating in the final $O_\perp$-phase structure. Conversely, Fig. 3h–j captures the $O_\perp$-*T* reverse transition. Figure 3i (intermediate state *IS-2*) clearly reveals that this state still retains a significant oxygen ion displacement characteristic similar to that of *O*-phase, while the lattice spacings of its alternating wide-narrow Hf/Zr sublattices gradually converge toward the lattice parameters of the tetragonal phase (2.47 Å/2.55 Å). This indicates that the $O_\perp$-*T* phase transition also follows a cation-first mechanism. The Hf/Zr cation sublattice undergoes distortion converging toward the intermediate state *IS-2*, followed by relaxation of oxygen ion displacements that revert the structure to the initial *T*-phase. Although both are intermediate states, *IS-1* and *IS-2* exhibit distinct alternating wide-narrow

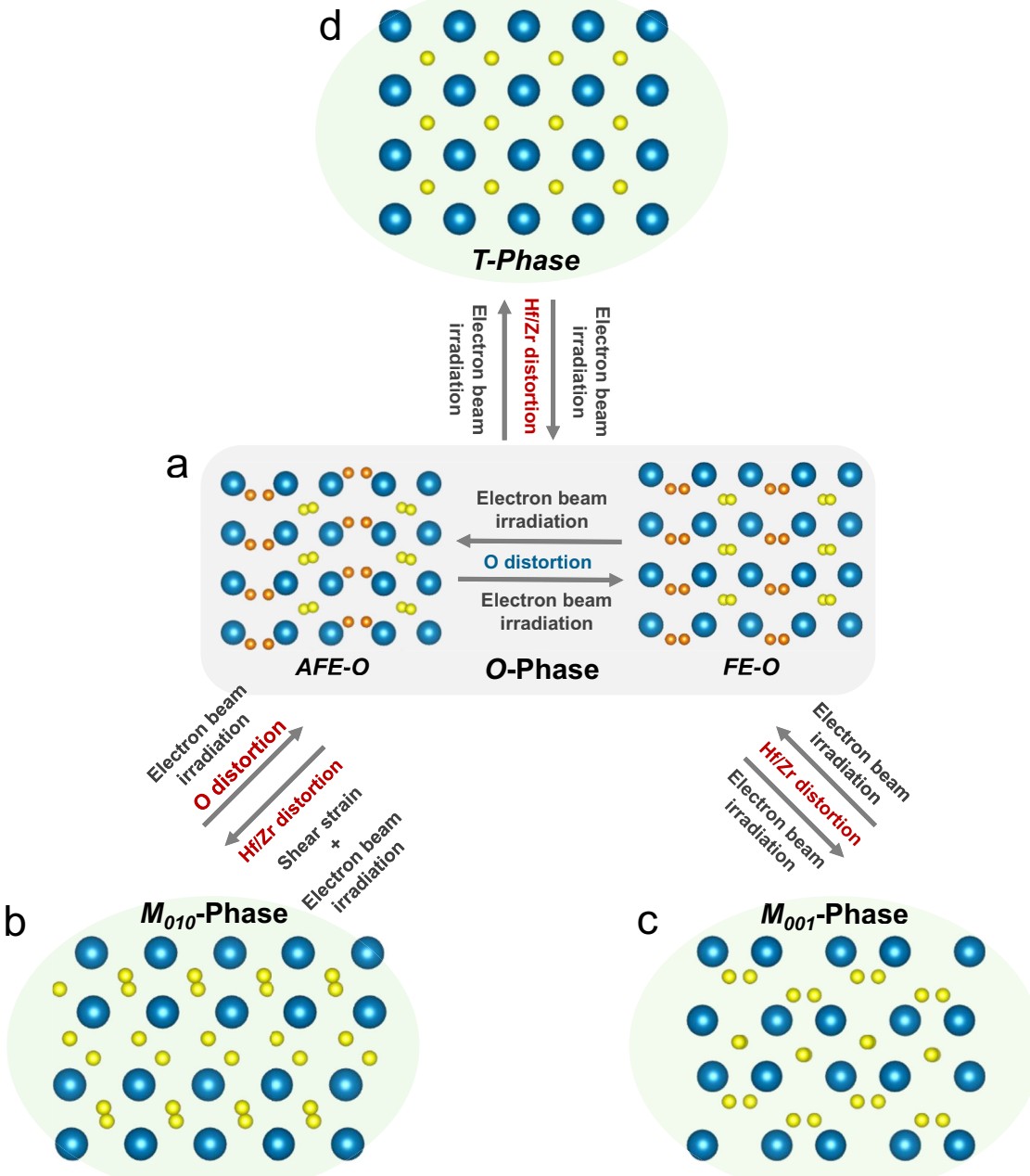

**Fig. 5 | Schematic of the phase transition behaviors in the superlattice film after electron beam irradiation. a** $O$-phase, including the $AFE$-$O$ and $FE$-$O$ phase. **b** $M_{O10}$-phase. **c** $M_{OO1}$-phase. **d** $T$-phase.

Hf/Zr sublattice spacings. This indicates that the reversible phase transition between *the T*-phase and $O_{\perp}$-phase proceeds via a multistep lattice-distortion pathway rather than via a single intermediate state, ultimately accomplishing the transformation between the polar $O_{\perp}$-phase and the nonpolar $T$-phase. These results elucidate a unified cation-sublattice-led distortion sequence governing reversible $T$-$O_{\perp}$ phase transitions in HfO$_2$-based ferroelectrics.

Comparing the phase transitions in Figs. 2a–f and 3 reveals a striking divergence. Despite both systems initiating from the $T$-phase, electron beam irradiation drives distinct transformation pathways, which contribute to the polarization orientation within the $O$-phase. In Fig. 3, the $O_{\perp}$-phase exhibits out-of-plane polarization and undergoes reversible transitions with the $T$-phase. Conversely, in Fig. 2a–f, the $O_{//}$-phase possesses in-plane polarization and progressively transforms into the $M_{OO1}$ phase under irradiation. The discrepancy of the phase

transition behaviors is attributed to the different strain states and interfacial effects within the $O_{\perp}$-phase and $O_{//}$-phase in the (HZO-ZO)$_6$ superlattice films. The $O_{\perp}$-phase distributes throughout both the HZO and ZO layers, exhibiting an in-plane normal strain modulation with an alternating 'contraction-expansion' character, and is unaffected by interfacial effects (Supplementary Fig. 14b). In contrast, the $O_{//}$-phase is constrained by the interface, primarily resides within the HZO layer, displays an out-of-plane normal strain modulation with an alternating 'contraction-expansion' character, and is accompanied by a significant out-of-plane lattice rotation (Supplementary Fig. 14c, e). The different strain states for the two types of $O$-phases are further confirmed by the quantitative strain analysis in Supplementary Fig. 15. Given the similarities and differences in atomic arrangements among the different phases, including the $O$-phase, $T$-phase, $M_{OO1}$-phase and $M_{O10}$-phase (Supplementary Figs. 16–17), strain states and interfacial effects

preferentially select the phase transformation pathways with higher structural compatibility and lower lattice distortion energy. Thereby, the $O_\perp$-phase tends to undergo reversible transitions with the $T$-phase (Fig. 3), whereas the $O_{//}$-phase favors transitions with the $M$-phase (Fig. 2). However, phase transition behaviors in the polycrystalline films are complicated by the complex interplay of high-density interfaces and defects. Notably, $O$-phases with different polarization orientations themselves interconvert under electron beam irradiation, as demonstrated in Supplementary Fig. 18. This directional dependence underscores polarization orientation as a critical determinant of phase stability and transformation kinetics in HfO$_2$-based ferroelectrics.

The $O$-phase discussed above encompasses both $FE$-$O$ and $AFE$-$O$ variants, both exhibiting the characteristic alternating wide/narrow Hf/Zr sublattice configuration (schematic in Fig. 4a, b). Despite their low energy gap[25] and common coexistence in thin films, they undergo distinct pathways during the reversible $T$-$O$ phase transition. When transforming from the $T$-phase to the $O_\perp$-phase, the $T$-phase preferentially transforms into the $FE$-$O_\perp$ phase rather than $AFE$-$O_\perp$ phase (yellow rectangle in Supplementary Fig. 13b), which is supported by previous theoretical predictions that the energy barrier for the transition between $T$-phase and $FE$-$O$ phase is lower than that for the transition between the $T$-phase and $AFE$-$O$ phase[35,36]. Conversely, during the transition from $O_\perp$-phase to $T$-phase, the $AFE$-$O_\perp$ phase first undergoes transformation into the $FE$-$O_\perp$ phase (red rectangle in Supplementary Fig. 13e), followed by its subsequent transformation into the $T$-phase (Supplementary Fig. 13f).

Furthermore, due to the minimal energy difference between $FE$-$O$ and $AFE$-$O$ phases[35,36], they can mutually transform under external stimuli, particularly during electron beam irradiation. Thus, real-time tracking of beam-induced structural evolution in HfO$_2$-based films facilitates to elucidate their phase-transition mechanisms. Figure 4c–f captures unit-cell-resolved $FE$-$AFE$ transitions at varying irradiation durations. Figure 4d captures the structural evolution following 10-minute electron-beam irradiation of the initial state (Fig. 4c), with arrows indicating polarization directions. This reveals unit-cell-scale polarization reversal in one HZO layer, resulting in a localized $AFE$-$O$ phase. Concurrently, attenuated ionic displacements in the bottom HZO ferroelectric layer accompany reduced out-of-plane Hf/Zr spacing, suggesting a potential metastable intermediate state during $O$-$T$ phase transformation. After 60-minute irradiation (Fig. 4e, f), coordinated polarization reversal across three layers stabilizes a global $AFE$-$O$ phase in HZO. In addition, reversible $AFE$-$FE$ transitions were also observed in this superlattice (Fig. 4g, h). Under 60-minute irradiation, initial coexistence of $AFE$-$O$ and $FE$-$O$ phases (Fig. 4g) evolves into a stabilized $FE$-$O$ phase with 180° domain walls via polarization reversal in the top HZO layer (Fig. 4h). More critically, under identical irradiation durations, the phase-transition volume is larger for the $FE$-to-$AFE$ transition (Fig. 4e, f) than for the reverse $AFE$-to-$FE$ process (Fig. 4g, h). This asymmetry provides direct experimental evidence for the lower energy state and greater stability of the $AFE$-$O$ phase, thereby validating earlier theoretical predictions[15,36]. Furthermore, Fig. 1h shows that the $O$-phase is laterally confined within the HZO layers, whereas the adjacent ZO layers maintain the $T$-phase structure. This originates from Zr-doping stabilizing $O$-phases in HfO$_2$, whereas pure ZrO$_2$ exhibits high $O$-phase energy barriers impeding ambient stability[16]. Strikingly, 60-minute electron irradiation stabilizes $AFE$-$O$ phases within ZO layers (Fig. 4i, j), demonstrating electron-beam irradiation as a viable pathway to achieve metastable $AFE$-$O$ states in undoped ZrO$_2$.

These findings illuminate the rich phase transformation mechanisms in HfO$_2$-based superlattice thin films under electron beam irradiation. The observed $O$-phase comprises both $FE$-$O$ and $AFE$-$O$ variants. Under irradiation, these phases undergo reversible switching, proceeding via initial O-ion distortion followed by Hf/Zr-ion distortion (Fig. 5a). Furthermore, the $O$-phase undergoes reversible transitions

into both the $T$-phase and the $M_{O01}$-phase, which are uniformly triggered by an initial distortion of the Hf/Zr ions (Fig. 5c, d). In contrast, the reversible transition between the $O$-phase and the $M_{O10}$-phase follows asymmetric ion-distortion pathways (Fig. 5b). The forward ($O$-to-$M_{O10}$) transformation requires the imposition of additional shear strain to initiate and is driven by Hf/Zr-ion distortion. Conversely, the reverse pathway ($M_{O10}$-to-$O$), initiates via O-ion distortion. Consequently, electron-irradiation-induced phase transformations in HfO$_2$-based superlattices follow specific ion-distortion sequences. These pathways are ultimately dictated by the symmetry discrepancy and ionic arrangement between the parent and product phases. It is worth noting that electrostatic charging is considered the primary mechanism for these phase transitions induced by electron beam irradiation, while the influence of localized heating is negligible under the low-dose conditions used in iDPC-STEM imaging[14,15]. This charging effect leads to a net positive charge accumulation on the surface of hafnia-based superlattice films, thereby generating an inhomogeneous electric field within the observed films. It is speculated that this emerging electric field could couple into the lattice, reducing the energy barrier of specific phase transition pathways and guiding the phase transition along a path analogous to that driven by an external electric field under device cycling. For example, the phase transition between $FE$-$O$ phase and $AFE$-$O$ phase observed in Fig. 4 is highly consistent with previous reports on HfO$_2$-based thin films under externally applied cycling voltages[25]. In a word, the reported rich phase transitions among $O$-phase and other non-polar phases under electron beam irradiation confirm the low energy barriers among these phases as reported previously[37], and further reveal the asynchronous sub-lattice distortion model governing these pathways. This model integrates the diverse and complex structural phase transition behaviors in HfO$_2$-based materials into a unified theoretical framework, systematically elucidating numerous controversies in prior research and providing a deeper, atomistic understanding of phase transition mechanisms.

As a result, this mechanistic insight into structural phase transitions provides a crucial experimental foundation for both device applications and structural design in HfO$_2$-based thin films. Firstly, it is revealed that the polarization switching in the $O$-phase requires only dipole-order reversal of the oxygen sublattice, without the need for large-scale lattice reconstruction. This mechanism demonstrates the inherent potential of HfO$_2$-based ferroelectric devices for achieving ultra-low switching energy, providing a theoretical basis for the design of low-power memory cells. Moreover, compared to mechanisms involving substantial cation displacements, the low-energy mechanism is likely more robust against thermal fluctuations, potentially enhancing device endurance at elevated temperatures. Secondly, the transition path from the $O$-phase to the $M$-phase is found to be highly dependent on the initial crystal orientation and projection direction. Given that the $M$-phase acts as a non-polar parasitic phase that degrades device performance, strategic orientation control during growth can suppress its formation or confine it to specific regions, even at high temperatures, thereby improving cycling endurance and retention in fabricated devices. Thirdly, the observed $FE$-$AFE$ reversibility and $O$-phase instability towards non-polar phases provide a microscopic basis for understanding device phenomena such as wakeup, fatigue, and endurance degradation. Therefore, these insights offer critical guidance for designing more reliable hafnia-based ferroelectric memories. Finally, it is found that the phase transition pathways respond selectively to strain states. By engineering interfacial stress and lattice mismatch in heterostructures, desired functional phases, such as the highly polar orthorhombic variant, can be stabilized at the nanoscale, effectively countering thermally induced phase instability. This level of control further enables the periodic arrangement of ferroelectric and antiferroelectric domains, paving the way for the creation of artificial superlattices with customized electronic properties.

This study focuses specifically on the mechanism of electron beam-induced structural phase transitions. While the timescales of electron beam irradiation-induced transitions are not directly comparable to the nanosecond-scale field-induced switching in devices, the atomistic pathways revealed here are expected to be fundamental. The asynchronous sublattice distortion mechanisms are governed by the intrinsic energy landscape of $HfO_2$-based materials, which also dictates the response under an electric field. Even though we fully recognize that other types of stimuli may lead to differences in phase transition types and kinetic processes. Further research on this basis, particularly exploring the influence of in-situ electric fields under different application modes, is of significant importance.

In summary, this work systematically reveals the rich spectrum of electron-beam-induced phase transitions in $HfO_2$-based superlattices and elucidates their distinct asynchronous sub-lattice distortion mechanisms. Reversible transitions between the $O$-phase and both the $T$-phase and $M_{OOI}$-phase are driven by the preferential distortions of the Hf/Zr sublattice, with the detailed phase transition pathways being dictated by the polarization direction of the $O$-phase. In contrast, the $O$-to-$M_{OIO}$ phase transformation requires additional shear strain and exhibits asymmetric ion-distortion modes during reversibility. Furthermore, polar-antipolar switching within the orthorhombic structure occurs solely via oxygen ion displacements. Collectively, this study provides a systematic elucidation of the intrinsic structural phase transition mechanisms in $HfO_2$-based thin films, establishing a fundamental framework for the design of high-performance hafnia-based ferroelectric electronic devices.

## Methods

### Film deposition details

Using pulsed laser deposition with a Coherent ComPex PRO 201 F KrF ($\lambda = 248$ nm) excimer laser, a series of $(Hf_{0.5}Zr_{0.5}O_2\text{-}ZrO_2)_6$ superlattice films were deposited on YSZ (001) substrates, including the $(Hf_{0.5}Zr_{0.5}O_2\text{-}ZrO_2\text{-}5)_6$, $(Hf_{0.5}Zr_{0.5}O_2\text{-}ZrO_2\text{-}6)_6$, and $(Hf_{0.5}Zr_{0.5}O_2\text{-}ZrO_2\text{-}11)_6$. The YSZ (001) substrates used here are commercial substrates without extra chemical or heat treatment. Before deposition, the substrates were heated to 900 °C for 20 min to clean the substrate surfaces and then cooled down to the film deposition temperature at a rate of 5 °C min⁻¹. The deposition of $Hf_{0.5}Zr_{0.5}O_2$ and $ZrO_2$ layers used the solid sintered $Hf_{0.5}Zr_{0.5}O_2$ and $ZrO_2$ targets, which was pre-sputtered for 20 min to clean the surface. When growing the $Hf_{0.5}Zr_{0.5}O_2$ and $ZrO_2$ layers, a repetition rate of 2 Hz, substrate temperature of 850 °C, oxygen partial pressure of 70 mTorr and laser energy of 2 J cm⁻² were used. After deposition, these films were annealed at 850 °C in an oxygen partial pressure of 200 Torr for 20 min and then cooled slowly to room temperature at a rate of 5 °C min⁻¹.

### XRD structural analysis

X-ray $\theta$-$2\theta$ scan and XRR scan were performed using a high-resolution X-ray diffraction with Cu Kα1 radiation in a AS-D100 diffractometer (Hefei APEX TECHNOLOGIES Co., Ltd.).

### TEM sample preparation, TEM observation

The cross-sectional TEM samples for STEM observation were prepared by a traditional process: slicing, gluing, grinding, dimpling, and finally ion milling. A Gatan 695 PIPS was used for ion milling. During the ion milling process, a low angle (5°) and a cooling stage were used first, and the final ion milling voltage was 0.3 eV for 10 min to reduce the beam damage. The selected area electron diffraction images were recorded using a conventional TEM (JEOL JEM-F200 working at 200 kV). HAADF-STEM, iDPC-STEM, and EDS-STEM images were recorded using Spectra 300 X-FEG aberration-corrected scanning transmission electron microscope (ThermoFisher Scientific) with double aberration (Cs) correctors and a monochromator operating at 300 kV. For the acquisition of HAADF-STEM and iDPC-STEM images, a spot size of 9 and a beam current of 100 pA were used; for EDS-STEM mapping, a spot size of 6 and a beam current of 100 pA were employed.

### STEM result analyses

Atom positions were accurately determined using 2D Gaussian peak fitting in Matlab[38], thus making it possible to acquire the information of the Hf/Zr- and O- atomic positions, Hf/Zr-ionic in-plane and out-of-plane rotation, and O-ionic displacement. It should be noted that a Wiener filter of HAADF and iDPC-STEM, and a low-pass annular mask restricted to the instrument resolution limit of the images, were used to reduce the noise of the obtained images. Quantitative stoichiometric analysis was conducted in Velox 3.14.0 using atomic-resolved EDS-STEM data, by applying the Empirical model for background correction and the Brown-Powell model for the ionization cross-section.

## Data availability

The data sets generated and analyzed during the current study are available from the corresponding author on reasonable request.

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

## Acknowledgements

This work is supported by National Natural Science Foundation of China (NO. 52571013 (W.R.G.), NO. 52201018 (W.R.G.), NO. 51971223 (Y.L.Z.), NO. 51901166 (S.R.Z.)), Guangdong Basic and Applied Basic Research Foundation (2024A1515140162 (W.R.G.), 2023A1515012796 (W.R.G.), 2021A1515110291 (W.R.G.)), Guangdong Provincial Quantum Science Strategic Initiative (GDZX2202001, GDZX2302001, GDZX2402001 (X.L.M.)), the Open Fund of the Microscopy Science and Technology, Songshan Lake Science City (202401202 (W.R.G.)). W.R.G. acknowledges the SLAB Young Scientists Program. We also acknowledge the XRD technical support provided by Y.Z. Chen at APEX TECHNOLOGIES.

## Author contributions

W.R.G. and B.R.W. contributed equally to this work. X.L.M. conceived the project on the architecture of quantum materials modulated by ferroelectric polarizations; W.R.G., Y.L.Z., and X.L.M. designed the sample structures and subsequent experiments. W.R.G. performed the thin-film growth and XRD observations. W.R.G. and B.R.W. performed the TEM and STEM observations. B.R.W. prepared the TEM samples. S.R.Z. and M.L. performed the digital analysis of the STEM data. All authors participated in the discussion and interpretation of the data.

## Competing interests

The authors declare no competing interests.
