## [Transparent Peer Review file · Nature Communications]

Roadmap of phase transitions in hafnia-based superlattice films

Corresponding Author: Professor Xiu-Liang Ma

Version 0:

Reviewer comments:

Reviewer #1

(Remarks to the Author)

This is an exceptionally interesting body of work describing the phase transformations in HZO superlattices by PLD and investigated using STEM. The phase transformations were induced by e-beam irradiation but it was assumed that these transformations were equivalent to those occurring in devices. I found the paper well thought-out and the experiments conducted in a thorough manner, it was extremely enlightening. I believe this warrants publication in Nature Communications. This work is significant in terms of progressing understanding in the field, not only in terms of fundamental understanding but also in engineering of a promising technology. However, I have some comments:

- Some of the figure labelling is wrong either in the text or on the figure. For example, Figure 5 and 6 seem to be swapped in the text compared to the labelling.
- When discussing "wide-narrow" sublattice patterns and the transformation routes, it is not clear which sublattice the authors are referring to, nor are the plots explained sufficiently. Please update the text to describe what is meant by sublattice in this case, label atoms in diagrams (it seems to have two colours of oxygen with no explanation), clarify which sublattice distances are being referred to, and explain 'Lattice X and Y' and clarify directions in the figures.
- The choice of using superlattice is not sufficiently explained. Why not use the more simple model of epitaxial HZO? Why choose these sublayer thicknesses? Why this periodicity? Clearly, the strain state in the HZO layers will vary with these parameters. A discussion as to the effects of superlattice parameters with respect to pure HZO films is needed. If the strain is modulated through the superlattice dimensions, this would surely have an effect on the phase transformation mechanics. A discussion here would clarify these phenomena.
- It remains unclear on the role of the e-beam irradiation and the mechanism in which it induces phase changes. Is it local heating? Electrostatic? It is claimed that these phase changes are the same as when induced by device cycling. It seems possible that different stimuli could lead to variation in the phase transformation type and mechanism. A discussion is needed to clarify this.
- There is some distinction about the difference in phase transformation mechanism depending on polarisation orientation. It is clear that differences in strain and interface effects in the two directions would likely have an effect on these changes. This is seen in the data when shear strain is added changing the transformation. Quantifying the strain in the in-plane and out-of-plane directions is important. Discussion should be added around the role of strain and interfaces. Also epitaxy vs polycrystalline film structures should be considered.
- Does the presence shear strain leading to O- to -M010 transformation mean that strain driven transformations are most likely to be O- to M-phase transformations?
- The authors state that by understanding mechanisms of phase transformations it would allow for the structural design of HfO₂-based materials. A small discussion around how this work is applicable to devices and route forward for structural design would add a lot to the value of this paper.
- It seems as though these time-dependent changes are very dynamic. What are the implications for device operation at elevated temperatures.
- In addition, there are some minor improvements to language and grammar needed.

Reviewer #2

(Remarks to the Author)

The authors report electron-beam-induced polymorphic phase transitions in HfO₂-based superlattices and introduce an asynchronous sublattice distortion mechanism. Transmission electron microscopy demonstrates that preferential sublattice distortions drive transitions among orthorhombic, tetragonal, and monoclinic phases. In addition, the polar-antipolar transition within the orthorhombic phase is attributed solely to the reversal of oxygen sublattice dipole order. The study is technically rigorous and supported by high-quality microscopy data. However, the novelty and broader significance of these findings remain uncertain, as outlined below:

-The electron-beam-induced phase transitions are intriguing, but ferroelectric devices are operated under applied electric fields, not electron irradiation. The work would gain much greater importance if the structural change mechanisms under voltage operation were investigated. What structural changes would the authors expect during electric-field experiments, and how might they differ from those observed under electron-beam excitation?

-While the authors propose a systematic framework for phase transitions, the unique contribution of the asynchronous sublattice distortion model compared to earlier experimental and theoretical studies is not sufficiently emphasized. What new insight does this model provide beyond existing reports?

-The rationale for using a superlattice structure is not entirely clear. Why are ZrO₂ layers included, and what is their functional role in the observed transitions? How would the results be expected to differ in a single HZO layer film without superlattice modulation?

-The manuscript reports facile phase transitions among the different polymorphs. Could the authors provide a discussion of the theoretical or experimental energy differences between these phases?

-The manuscript frequently uses the phrase "It is noteworthy that" which appears repetitive. The authors should revise such instances to improve readability and stylistic variety.

Reviewer #3

(Remarks to the Author)

The paper presents an investigation into intriguing structural phase transitions of HfO₂-based superlattice thin films using in-situ aberration-corrected STEM under electron beam irradiation. Asynchronous sublattice distortion mechanisms seemed to be interesting. This proposal is a timely topic in hafnia ferroelectricity and worthwhile to be reported.

In my opinion, this study demonstrated clear experimental results to be suited for Nature Communications and might be publishable. However, my general feeling is that the manuscript could be better written with more concise physical explanations and clear experimental details. In the manuscript, there is little attempt in explaining how to characterise and understand the intermediate states and the relation between irradiation and electric field-driven switching in the real devices. To be considered for publication, I suggest some major revisions to be made. The followings have been noted and need to be addressed.

- The proposed mechanism of asynchronous sublattice distortion, distinguishing Hf/Zr-initiated versus O-initiated phase transitions, is both novel and compelling. This conceptual framework can have broad implications beyond HfO₂-based materials, possibly informing studies on ZrO₂, TiO₂, and other oxide-based functional materials. The use of epitaxial superlattices as a model system is particularly commendable, as it effectively isolates the intrinsic transformation behavior from extrinsic effects such as grain boundaries and interfacial disorder. For the general readers, the discussion could benefit from a clearer comparison to previously proposed phase transition models, particularly those involving field-induced switching and thermodynamic simulations.

-The observation of reversible FE–AFE transitions and T–O phase reversibility is highly relevant to device reliability. It would be helpful to quantify or discuss the kinetics and energy barriers of the observed transitions, at least qualitatively. (1) How do the observed phase transition timescales under e-beam irradiation compare with those under electric field cycling in actual devices? (2) Are these pathways accessible under field-induced switching, or are they artifacts of the beam–matter interaction? (3) Could the authors comment on how closely the electron-beam-induced pathways mimic the actual operational phase transitions in memory applications?

- The XRD data shown in Fig. 1g appears to be limited to a narrow angular range near the (002) reflection. For a more comprehensive understanding of the superlattice structure and to verify the presence of (001) reflection and other reflections related to the superlattice, I suggest that the authors provide an extended scan that includes the (001) region as well. If available, please also provide x-ray reflectivity. This would help confirm whether any superlattice-related periodicity or structural modulation is detectable in reciprocal space.

Experimental details

- If available, please provide the P-E hysteresis curves of pristine HZO samples in comparison with those of the irradiated counterparts to better illustrate the effect of electron beam exposure on ferroelectric switching.
- Provide experimental evidence or details regarding how the cation/anion stoichiometry was characterised in the HZO superlattice samples.

Version 1:

Reviewer comments:

Reviewer #2

(Remarks to the Author)

The authors have responded to all comments thoroughly and revised the manuscript accordingly. Therefore, I recommend this manuscript for publication.

Reviewer #3

(Remarks to the Author)

The referee recognizes the authors' efforts in addressing most of the comments and appreciates the expanded discussion of the asynchronous sublattice distortion model. The additional reference, revised figures, and more detailed explanations improve the clarity and depth of the manuscript.

However, the referee still has important concerns, primarily related to the structural characterisation and the lack of direct ferroelectric evidence. These points are critical to validate the manuscript's central claims and need major revision prior to consideration for publishing in Nature Communications.

(1) The authors claim that the film is "a (Hf_{0.5}Zr_{0.5}O₂)₆ superlattice with a period of 7 unit cells per layer on a [001]-oriented YSZ substrate" in the revised manuscript, as opposed to 6 unit cells. Revised Supplementary Fig. S7, however, presents a different film with a 5 unit-cell periodicity, resulting in a discrepancy in the sample description. The referee recommends that this ambiguity be definitively addressed at the beginning of the manuscript.

Furthermore, the XRD and XRR data currently shown are not sufficient to verify superlattice periodicity and structural modulation. While the authors provide XRR data above 0.6°, the signal only shows a single broad oscillation, without distinct superlattice reflection peaks. In comparison, previous studies [refer to Li et al., Nat. Comm. 16, 6417 (2025)] demonstrate well-resolved superlattice peaks in XRR.

The referee recommends that the authors conduct synchrotron-based XRD and XRR studies to resolve the superlattice peaks and enhance the signal-to-noise ratio. This would help whether there are detectable structural modulations or symmetry changes after e-beam irradiation.

This data is essential to examine if structural modulation in the forbidden (001) reflection region, where substrate signal is suppressed, indicates any modifications before and after electron beam irradiation, which is pivotal to the manuscript's claims. Given that the authors report clear asymmetric sublattice distortions and complex phase transitions at the atomic scale via STEM, it is crucial to figure out if these structural modifications also lead to measurable signals in reciprocal space. If such distortions are intrinsic and widespread inside the irradiated volume, they should appear as modulations in the XRD pattern, particularly in areas of reciprocal space that are often background-suppressed. See also Solomon et al., Sci. Adv. 11, eadq5943 (2025) for an example where beam-induced structural changes in zirconia were clearly evident in XRD analyses.

(2) The referee agrees with the authors' explanation that it was not possible to measure P-E hysteresis because there was a lot of leakage current in the superlattice structure with in-plane interdigitated electrodes. While the experimental challenge is understandable, this constraint is especially significant due to the manuscript's focus on structural phase transitions closely linked to the presence of polar or ferroelectric phases.

In HfO₂-based systems, reliable P-E measurements have been shown even in ultrathin films, typically due to their large coercive fields. The inability to evaluate switching characteristics in the current samples raises concerns regarding material quality or structural defects, particularly given that the proposed phase transitions involve noticeable structural distortions and the emergence of potential defective states.

The referee recommends that the authors provide current density-voltage (J-V) or leakage current data measured in the same in-plane interdigitated configuration. Even in the absence of a measurable switching signal, such data would allow readers to independently assess the leakage behaviour and evaluate its implications for the functional quality of the film, particularly given the structural distortions reported under electron beam irradiation.

As a complementary approach, the authors can consider techniques such as second harmonic generation (SHG) to probe the presence of polar orders.

The manuscript addresses a significant and timely topic, and the concept of asynchronous sublattice distortion is intriguing. Nonetheless, the structural evidence and ferroelectric measurements need more experimental validation, particularly if this work is to be regarded as a definitive reference in the field.

The referee recommends major revision and looks forward to reviewing a strengthened version of the manuscript.

Version 2:

Reviewer comments:

Reviewer #3

(Remarks to the Author)

After evaluating the revisions made by the authors in response to the reviewer's comments, the referee finds that most aspects of the study have been adequately addressed. In particular, the author performed additional X-ray diffraction measurements and second harmonic generation experiments to more carefully analyse the structural properties of the superlattice films. These measurements provide clear signatures in the superlattice samples. Overall, the authors have made a genuine effort to address the previous comments and questions that are important for readers to better understand the claims presented in this work. The manuscript is well suited for publication in a journal with the scope of Nature Communications.

Reply to referees' questions and comments:

Ref number: NCOMMS-25-68598

Title: Roadmap of phase transitions in hafnia-based superlattice films

Authors: Wan-Rong Geng, Bo-Rui Wang, Yin-Lian Zhu, Si-Rui Zhang, Min Liao, Xiu-Liang Ma

16 December, 2025

Reply to Reviewer #1 (R1):

We appreciate the positive comment by the referee that “This is an exceptionally interesting body of work describing the phase transformations in HZO superlattices by PLD and investigated using STEM”, “I found the paper well thought-out and the experiments conducted in a thorough manner, it was extremely enlightening. I believe this warrants publication in Nature Communications” and “This work is significant in terms of progressing understanding in the field, not only in terms of fundamental understanding but also in engineering of a promising technology”.

In the meanwhile, the referee also raises several specific questions and comments which are summarized into nine aspects. We fully understand the referee’s concerns, and here we have addressed all the questions and discussed all the comments one by one in the following. The revisions are written in **RED** in the revised manuscript and the revised Supplementary Materials.

Question and comment (R1.1): Some of the figure labelling is wrong either in the text or on the figure. For example, Figure 5 and 6 seem to be swapped in the text compared to the labelling.

Reply to Question and comment (R1.1):

We appreciate the kind reminder from the referee. As suggested by the referee, we have corrected the figure labelling in the revised Supplementary Materials.

Question and comment (R1.2): When discussing "wide-narrow" sublattice patterns and the transformation routes, it is not clear which sublattice the authors are referring

to, nor are the plots explained sufficiently. Please update the text to describe what is meant by sublattice in this case, label atoms in diagrams (it seems to have two colours of oxygen with no explanation), clarify which sublattice distances are being referred to, and explain 'Lattice X and Y' and clarify directions in the figures.

Reply to Question and comment (R1.2):

We appreciate the concerns and suggestions by the referee. In the revised Supplementary Materials, we have updated the Supplementary Fig. 4, where the ‘wide’ and ‘narrow’ Hf/Zr sublattices in the *AFE O*-phase and *FE O*-phase of the HZO are defined. The ‘wide’ and ‘narrow’ sublattices correspond to the ferroelectric layer and spacer layer of the *O*-phase HZO, respectively¹. Thus, two kinds of oxygen atoms are highlighted, including the O₁ in the ferroelectric layer and O₂ in the spacer layer. Besides, the in-plane lattice constant (Lattice X) and out-of-plane lattice constant (Lattice Y) for different phases of the HZO are also defined in Supplementary Fig. 4.

Supplementary Fig. 4 | Multiple polymorphic phases in hafnium-based films. a, M-phase. b, T-phase. c, AFE O-phase. d, FE O-phase. The “wide” and “narrow” Hf/Zr sublattices are highlighted in the *AFE O*-phase and *FE O*-phase. **e, Definition of the Lattice X and Lattice Y for different phases.** Blue, orange and yellow balls denoting the Hf/Zr, O₁ and O₂ atomic columns, with the O₁ and O₂ representing the oxygen atoms in ferroelectric layer and spacer layer, respectively.

Furthermore, in the revised Manuscript, we have also added some discussions to

illuminate the ‘one-wide-one-narrow’ Hf/Zr sublattice characteristics of the *O*-phase HZO and highlighted the Hf/Zr sublattice through the whole Manuscript.

Lines 1-9, Page 6: “As the crucial *O*-phase in the hafnia-based materials, the polar and anti-polar *O*-phases display the distinct structural characteristics of ‘one-wide-one narrow’ Hf/Zr sublattice distribution (Supplementary Fig. 4) accompanied with the non-centrosymmetric displacement of oxygen atoms, with the ‘wide’ sublattice being the ferroelectric layer and ‘narrow’ sublattice being the spacer layer. Thus, two kinds of oxygen atoms are also denoted, including the O_1 in the ferroelectric layer and O_2 in the spacer layer, as shown in Supplementary Fig. 4. The evident structural features play the vital role on determining the phase transition between *O*-phase and other nonpolar phases in the hafnia-based materials. Complementary analysis in Fig. 2 elucidates the transition mechanisms between the *O*-phase and *M*-phase (Supplementary Fig. 9), which is crucial for understanding and optimizing the ferroelectric switching behavior of HfO₂-based films, and is pivotal in addressing device reliability and performance degradation issues.”

Question and comment (R1.3): The choice of using superlattice is not sufficiently explained. Why not use the more simple model of epitaxial HZO? Why choose these sublayer thicknesses? Why this periodicity? Clearly, the strain state in the HZO layers will vary with these parameters. A discussion as to the effects of superlattice parameters with respect to pure HZO films is needed. If the strain is modulated through the superlattice dimensions, this would surely have an effect on the phase transformation mechanics. A discussion here would clarify these phenomena.

Reply to Question and comment (R1.3):

We appreciate the constructive discussions and good proposals from the referee. As noticed by the referee, we illuminated the detailed phase transition behaviors of the (HZO-ZO)₆ epitaxial superlattice thin film under electron-beam irradiation. In hafnia-based materials, the ferroelectricity primarily originates from the metastable orthorhombic (*O*) phase (*Pca2₁*). Therefore, stabilizing this polar *O*-phase and understanding its phase transitions to non-polar phases is critical. This fundamental understanding is essential for enhancing the macroscopic ferroelectric properties of hafnia-based materials and advancing their commercial applications. To eliminate the interference from high-density defects and interfaces in polycrystalline films to the analysis of phase transition mechanisms, it is necessary to construct a simpler single-crystal film system. Thus, the hafnia-based films are epitaxially grown on YSZ (001) substrates, considering the fact that the YSZ exhibits similar structural symmetry and a

high degree of lattice matching with HZO, ZrO₂, and HfO₂. This compatibility enables the growth of high-quality, epitaxial single-crystal films, thereby avoiding interference from the complex microstructure of polycrystalline films in the analysis of phase transition mechanisms.

On the other hand, to elucidate the structural phase transition mechanisms between the *O*-phase and other non-polar phases in hafnia-based films, it is first necessary to construct a film system with the metastable *O*-phase being stabilized. Although a single-layer HZO film has a simpler structure, preliminary studies have shown that single-layer HZO single-crystal films epitaxially grown on YSZ (001) substrates are predominantly composed of the non-polar monoclinic (*M*) phase ($P2_1/c$), making it difficult to stabilize the metastable orthorhombic ferroelectric phase in such films (Fig. R1.1). Superlattice thin-film systems, as typical size-confined systems, are constructed by periodically alternating the layer growth of different materials, introducing significant chemical, strain, symmetry, charge, and polarization discontinuities between layers²⁻⁴. These result in strong interfacial and interlayer interactions. Consequently, they hold the potential to overcome the physical limitations of bulk materials, stabilize new structures, and induce novel physical properties comparing with the pure HZO films. Thereby, we constructed a (HZO-ZO)₆ superlattice single-crystal film system, successfully stabilized the polar *O*-phase, and further systematically investigated the phase transition behaviors and structural mechanisms between the *O*-phase and other phases using electron-beam irradiation experiments in an aberration-corrected transmission electron microscopy.

Fig. R1.1| Atomic-resolved HAADF-STEM image of the 10 nm Hf_{0.5}Zr_{0.5}O₂ film, with the nonpolar monoclinic phase ($P2_1/c$) being stabilized.

In this study, we systematically investigated the phase transition behaviors in the

(HZO-ZO-7)₆ superlattice films with the growth period thickness of 7 unit-cells under electron beam irradiation. The as-deposited films initially exhibit a coexistence of *O*-, tetragonal (*T*), and *M*-phases, with the *O*-phase featuring both in-plane (*O*_{//}) and out-of-plane (*O*_⊥) polarization orientations. The structural transition pathways and underlying mechanisms among *T*-*O*-*M*₀₀₁, *O*-*M*₀₁₀, *T*-*O*, and ferroelectric-antiferroelectric *O*-phase (FE-*O*-AFE-*O*) are then revealed. It is noteworthy that in the (HZO-ZO) superlattice system, the number of periods exhibits less impact on the phase structure, while the growth period thickness plays the crucial role on modulating the phase composition of the superlattice films. As suggested by the referee, we fabricated three types of (HZO-ZO)₆ superlattices with varying period thicknesses, including (HZO-ZO-5)₆, (HZO-ZO-7)₆, and (HZO-ZO-11)₆ superlattice films. Atomic-scale aberration-corrected transmission electron microscopy studies revealed distinct phase configurations among them. The (HZO-ZO-5)₆ film displays the uniform structure of *T*-phase without the existence of *O*-phase (Fig. R1.2). The (HZO-ZO-7)₆ film displays the multi-phase coexistence as discussed in the present Manuscript (Fig. R1.3). However, despite sharing a multiphase coexistence, the (HZO-ZO-11)₆ superlattice film shows the layered phase structure. The first two HZO layers adopt the *M*-phase oriented along different zone axes. The third layer exhibits the *O*_{//}-phase and the fourth layer displays the coexistence of *O*_{//}-phase and *O*_⊥-phase, then the fifth layer stabilizes the *O*_⊥-phase (Fig. R1.4).

However, with increasing the growth period thickness, the growth quality of the superlattice films progressively deteriorates, accompanied by enhanced interfacial curvature between layers. Notably, in the near-surface region of the (HZO-ZO-11)₆ film, partial interdiffusion of HZO into ZO layers is observed. Therefore, based on a comprehensive evaluation of structural quality and phase complexity, the (HZO-ZO-7)₆ superlattice was selected as the primary system for in-depth observation and analysis in this study.

Fig. R1.2| Atomic-resolved HAADF-STEM image of the (HZO-ZO-5)₆ superlattice

film, with the thickness of one (HZO-ZO) period being 5 unit-cells.

Fig. R1.3| Atomic-resolved HAADF-STEM image of the (HZO-ZO-7)₆ superlattice film, with the thickness of one (HZO-ZO) period being 7 unit-cells.

Fig. R1.4| Atomic-resolved HAADF-STEM image of the (HZO-ZO-11)₆ superlattice film, with the thickness of one (HZO-ZO) period being 11 unit-cells.

As suggested by the referee, we further performed the strain analysis of the three types of (HZO-ZO)₆ superlattice films. In Fig. R1.5a-c, the strain states of ϵ_{xx} , ϵ_{yy} , R_y and R_x are defined. By comparing the different strain states in the three types of

superlattice films, the evident differences derive from the different phase structures. For the (HZO-ZO-5)₆ superlattice film, the uniform strain states are displayed through the entire superlattice film, considering the single *T*-phase in the superlattice film. With the thickness of one (HZO-ZO) growth period increasing to 7 unit-cells, the whole superlattice film also presents the uniform states of ϵ_{xx} and R_x , besides the horizontally stripe-like strain fluctuation in the maps of R_y and ϵ_{yy} derived from the *O*_{//}-phase. When the thickness of one (HZO-ZO) growth period is further increase to 11 unit-cells, evident strain discrepancy is observed in R_y , R_x and ϵ_{yy} map, while the ϵ_{xx} map still presents the uniform strain distribution. It is noted that the obvious strain distinction attributes to the coexistence of multiple phases. In a word, the strain states in the superlattice films would not be modulated through the change of superlattice dimensions, but by the stabilized phase structures with different symmetry. Thus, it is proposed that the phase transitions among different phases share a common structural mechanism in the (HZO-ZO)-typed superlattice films.

Fig. R1.5| Strain analysis of the (HZO-ZO)₆ superlattice films with different thickness of one (HZO-ZO) period. a, Schematic of the in-plane normal strain (ϵ_{xx}). **b**, Schematic of the out-of-plane normal strain (ϵ_{yy}). **c**, Schematic of the in-plane lattice rotation (R_x) and out-of-plane lattice rotation (R_y). **d-g**, Strain analysis of ϵ_{xx} , R_y , R_x and ϵ_{yy} of the (HZO-ZO-5)₆ superlattice film. **h-k**, Strain analysis of ϵ_{xx} , R_y , R_x and ϵ_{yy} of the (HZO-ZO-7)₆ superlattice film. **l-o**, Strain analysis of ϵ_{xx} , R_y , R_x and ϵ_{yy} of the (HZO-ZO-11)₆ superlattice film.

Changes in the revised Manuscript and revised Supplementary Materials:

To illuminate the reasons for using a superlattice structure, we have added some discussions in Lines 16-21, Page 4 in the revised Manuscript.

Lines 16-21, Page 4: “Given that $\text{Y}_2\text{O}_3\text{:ZrO}_2$ (YSZ) substrates share the fluorite structure with HfO_2 - and ZrO_2 -based materials, the epitaxial growth of HfO_2 -based thin films on YSZ substrates enables superior crystalline quality. Furthermore, superlattice thin-film systems, as typical size-confined systems, are constructed by periodically alternating the layer growth of different materials, displaying the strong interfacial and interlayer interactions. They are expected to overcome the physical limitations of bulk materials, for example, stabilizing the desired polar structures and inducing novel physical properties in HfO_2 -based materials^{27,28}.”

Besides, the Fig. R1.2 and Fig. R1.4 have been added as Supplementary Fig. 7 and Supplementary Fig. 8 in the revised Supplementary Materials. Some sentences have also been added in Lines 18-21, Page 5 in the revised Manuscript.

Lines 20-23, Page 5: “The *O*-phase exhibits two distinct orientations: (i) horizontally aligned domains predominantly within the HZO layer (*O*_{//}-phase, pink mask), with the adjacent ZO layer maintaining a nonpolar *T*-phase structure; (ii) vertically oriented domains traversing both HZO and ZO layers (*O*_⊥-phase, cyan mask). It is revealed that the phase structure, spatial distribution, and epitaxial quality of the superlattice films depend on the thickness of one (HZO-ZO) period (Supplementary Fig. 7-8). The particular (HZO-ZO)₆ film investigated here, with a period thickness of 7 unit-cells, exhibits both high growth quality and a stabilized polar *O*-phase.”

Question and comment (R1.4): It remains unclear on the role of the e-beam irradiation and the mechanism in which it induces phase changes. Is it local heating? Electrostatic? It is claimed that these phase changes are the same as when induced by device cycling. It seems possible that different stimuli could lead to variation in the phase transformation type and mechanism. A discussion is needed to clarify this.

Reply to Question and comment (R1.4):

We appreciate the constructive discussions and good questions from the referee. As noticed by the referee, we have illuminated the phase transitions in hafnia-based superlattice films under electron-beam irradiation.

The electron beam irradiation is a powerful tool to regulate the sample structures and the physical properties, which has been discussed in many previous reports⁵⁻⁸.

However, the interaction between high-energy electrons and specimens in a transmission electron microscope is very complicated, and the irradiation damage mechanisms can be classified into atomic displacement, surface sputtering, radiolysis, electrostatic charging, and heating^{9,10}. During the iDPC-STEM imaging of the fluorite oxide-based materials, the influence of atomic displacement and heating have been ruled out due to the low-dose imaging conditions^{11,12}. Surface sputtering and radiolysis could introduce localized oxygen vacancies on materials' surface that might contribute to the switching and phase transition processes, but their influences are much smaller than that of electrostatic charging¹¹.

The charging effect is pronounced in electrically insulating materials^{6,10,12}, particularly in HfO₂, ZrO₂ and HZO owing to their large bandgap. In our experiment, the scanning electron beam causes the emission of secondary and Auger electrons, thereby generating a net positive charge on the surface of hafnia-based superlattice films that persists due to insufficient charge dissipation through conduction. Thus, the charging effect induced by the scanning beam can create an inhomogeneous electric field^{6,11,12} in the observed hafnia-based superlattice films. The electric field not only induces phase transition inside the superlattice films, but also induces stress field due to the changes of lattice constants and symmetry of different phases. However, it is noted that the rapid scanning of a fine electron probe across a large area makes the direction of the induced electric field difficult to define.

We fully agree with the referee that different stimuli could lead to variation in the phase transformation type and mechanism. As discussed above, electron beam irradiation induces the accumulation of positive charges within the insulating hafnia-based superlattice films, leading to the formation of a strong built-in electric field inside the thin films. This field may produce the effects similar to those of the external electric field applied via electrodes during device cycling. Therefore, it is speculated that the built-in electric field generated by the electron beam can also couple into the lattice system, reducing the energy barrier of specific phase transition pathways and guiding the phase transition along a path analogous to that driven by an external electric field.

For instance, the phase transition between *FE-O* phase and *AFE-O* phase observed in Fig. 4 in this Manuscript is highly consistent with previous reports on HfO₂-based thin films under externally applied cycling voltages¹³. This consistency of phase transition behaviors suggests that, despite the different physical stimuli, the material system is driven into a similar free energy landscape, thereby following comparable phase transition pathways. Thus, whether it is the external electric field in devices or the built-in electric field induced by the electron beam, both ultimately act on the ions in the crystal lattice. Both may overcome energy barriers through electrostatic

interactions, promoting ion displacement and ultimately achieving the phase transition. Based on this, electron beam irradiation can be regarded as a localized, electrode-free "electric field injection" method.

Changes in the revised Manuscript:

To illuminate the role of the e-beam irradiation in inducing the phase transitions and its relationships with that of cycling fields, we have added some discussions in Lines 25-30, Page 12, Lines 1-8, Page 13 and Lines 11-20, Page 14 in the revised Manuscript.

Lines 25-30, Page 12 and Lines 1-8, Page 13: “Consequently, electron-irradiation-induced phase transformations in HfO₂-based superlattices follow specific ion-distortion sequences. These pathways are ultimately dictated by the symmetry discrepancy and ionic arrangement between the parent and product phases. It is worthwhile to note that the electrostatic charging effect is considered the primary mechanism for these phase transitions induced by electron beam irradiation, while the influence of localized heating can be considered negligible under the low-dose conditions used in iDPC-STEM imaging^{13,14}. This charging effect leads to a net positive charge accumulation on the surface of hafnia-based superlattice films, thereby generating an inhomogeneous electric field within the observed films. It is speculated that this emerging electric field could couple into the lattice, reducing the energy barrier of specific phase transition pathways and guiding the phase transition along a path analogous to that driven by an external electric field under device cycling. For example, the phase transition between *FE-O* phase and *AFE-O* phase observed in Fig. 4 is highly consistent with previous reports on HfO₂-based thin films under externally applied cycling voltages²⁴.”

Lines 11-20, Page 14: “It should be noted that this study focuses specifically on the mechanism of electron beam irradiation-induced structural phase transitions. While the timescales of electron beam irradiation-induced transitions are not directly comparable to the nanosecond-scale field-induced switching in devices, the atomistic pathways revealed here are expected to be fundamental. The asynchronous sublattice distortion mechanisms are governed by the intrinsic energy landscape of HfO₂-based materials, which also dictates the response under an electric field. Even though, we fully recognize that other types of stimuli may lead to differences in phase transition

types and kinetic processes. Further research on this basis, particularly exploring the influence of *in-situ* electric fields under different application modes, is of significant importance.”

Question and comment (R1.5): There is some distinction about the difference in phase transformation mechanism depending on polarisation orientation. It is clear that differences in strain and interface effects in the two directions would likely have an effect on these changes. This is seen in the data when shear strain is added changing the transformation. Quantifying the strain in the in-plane and out-of-plane directions is important. Discussion should be added around the role of strain and interfaces. Also epitaxy vs polycrystalline film structures should be considered.

Reply to Question and comment (R1.5):

We appreciate the constructive suggestions from the referee. As noticed by the referee, there is the distinct differences in the mechanisms of phase transition depending on the polarization orientation of the *O*-phase. We fully agree with the referee that the strain states and interface effects in the two directions are different. As suggested by the referee, we have provided the detailed strain distribution of the superlattice film including the horizontal-polarized orthorhombic phase (*O*_{//}-phase) and vertical-polarized orthorhombic phase (*O*_⊥-phase) in Fig. R1.6. The *O*_⊥-phase presents the alternative “expansion-contraction” modulation in the in-plane normal strain (ϵ_{xx}) map, as highlighted by the region 1 in Fig. R1.6b, while no lattice modulation in the out-of-plane normal strain (ϵ_{yy}) map (region 3 in Fig. R1.6c). In contrast, the *O*_{//}-phase phase in the HZO layers displays the similar lattice modulation in the ϵ_{yy} map (region 4 in Fig. R1.6c) and no lattice distortion in the ϵ_{xx} map (region 2 in Fig. R1.6b).

Besides, the interface effects for the two types of the *O*-phases also presents evident discrepancies. For the *O*_⊥-phase, the in-plane lattice rotation (R_x) map and out-of-plane lattice rotation (R_y) map display the nearly uniform contrast, suggesting no interface effects. It attributes to the *O*-phase of this orientation forming a continuous structure across the interface, distributed within both the HZO and ZO layers, thereby suppressing any significant interface effects. However, the case is different for the *O*_{//}-phase. Despite the uniform contrast in the R_x map, the interfaces between HZO layers (*O*-phase) and ZO layers (*T*-phase) display distinct shear strain discrepancy in the R_y map (Fig. R1.6e), suggesting the interface effects.

Fig. R1.6| Detailed strain distribution for the orthorhombic phase. a, Atomic-resolved iDPC-STEM image including both the O_{\perp} -phase and $O_{//}$ -phase. **b,** In-plane normal strain (ϵ_{xx}) map. **c,** Out-of-plane normal strain (ϵ_{yy}) map. **d,** In-plane lattice rotation (R_x) map. **e,** Out-of-plane lattice rotation (R_y) map.

To quantitatively analyze the strain states within the O -phase along the two orientations, the strain distribution profiles for the four regions in Fig. R1.6 are also presented in Fig. R1.7. For the O_{\perp} -phase in region 1, the average in-plane normal strain ranges from -8.3 to 7.2 (Fig. R1.7a). However, the average out-of-plane normal strain of the $O_{//}$ -phase in region 4 fluctuates between the two values of -7.1 and 6.7 (Fig. R1.7d). The strain distribution of ϵ_{xx} in region 2 and ϵ_{yy} in region 3 present the nearly uniform values (Fig. R1.7b-c).

Fig. R1.7| Quantitative strain analysis. **a**, The ϵ_{xx} spacing profile for the region 1 along in-plane direction. **b**, The ϵ_{xx} spacing profile for the region 2 along in-plane direction. **c**, The ϵ_{yy} spacing profile for the region 3 along out-of-plane direction. **d**, The ϵ_{yy} spacing profile for the region 4 along out-of-plane direction.

As demonstrated in this study, O -phases with different polarization directions could lead to distinct phase transition behaviors. Specifically, the O_{\perp} -phase tends to undergo reversible structural transitions with the non-polar tetragonal phase, while the $O_{//}$ -phase preferentially exhibits reversible transformations with the M -phase. Moreover, the introduction of shear strain associated with defects further promotes the transformation of the O -phase into the M_{010} -phase. The differences of phase transition behaviors can be attributed to the differences of strain states and interfacial effects.

It is noted that the O_{\perp} -phase and T -phase tend to present across the interfaces within the HZO and ZO layers in the superlattice film. Thus, the interfacial effects have less influence on their stability and structural transitions to some extent. Detailed lattice analysis reveals that the O_{\perp} -phase exhibits an ‘one-wide-one-narrow’ distribution of square Hf/Zr sublattices in the in-plane direction, accompanied by significant oxygen-ionic displacements at the sub-unit-cell scale (Fig. R1.8b). Similarly, the T -phase displays the square Hf/Zr sublattices but without the oxygen-ionic displacements (Fig. R1.8a). Therefore, the structural transition between the O_{\perp} -phase and T -phase could be realized through these steps, including in-plane contraction/expansion of the Hf/Zr sublattices and shifts of oxygen ions, as illustrated in Fig. 3 in the Manuscript. Consequently, reversible phase transitions are expected to occur between the O_{\perp} -phase and the T -phase under electron beam irradiation.

Fig. R1.8| Unit cell schematics of T -phase (a) and O_{\perp} -phase (b).

Fig. R1.9| Unit cell schematics of $O_{//}$ -phase (a), M_{001} -phase (b) and M_{010} -phase (c).

On the other hand, the phase transition behavior between the $O_{//}$ -phase and the M -phase exhibits distinct structural characteristics. The unit cell structures of the M -phase differ significantly along various projection directions. As shown in Fig. R1.9a, the $O_{//}$ -phase displays a ‘one-wide-one-narrow’ square arrangement of Hf/Zr sublattices along with significant oxygen-ionic displacements. In contrast, the unit cell of the M_{001} -phase consists of alternating upright and inverted trapezoidal structures along the in-plane and out-of-plane directions, also exhibiting a ‘one-wide-one-narrow’ distribution of Hf/Zr sublattices in the out-of-plane direction (Fig. R1.9b). Therefore, the structural transition between the $O_{//}$ -phase and the M_{001} -phase requires coordinated lattice distortions involving the horizontal movement of Hf/Zr atomic columns and oxygen-ionic displacements, as illustrated in Fig. 2a-i in the Manuscript.

However, the M_{010} -phase exhibits unique structural characteristics. When viewed along the $[010]$ zone axis, the monoclinic angle of this phase is clearly observable. Its unit cell is composed of a combination of parallelogram and square Hf/Zr sublattices (Fig. R1.9c). Consequently, the transition from the $O_{//}$ -phase to the M_{010} -phase requires the stretching of the parallelogram-shaped Hf/Zr sub-lattices into the square grids, a process that necessitates additional shear strain as the driving force, as demonstrated in Fig. 2g-o in the Manuscript.

Furthermore, it is important to emphasize that the interfacial effects play the crucial role during the structural transition between the $O_{//}$ - and M -phases. In the as-grown $(\text{HZO-ZO})_6$ superlattice thin film, the $O_{//}$ -phase tends to reside in the HZO layers rather than in the ZO layers. Therefore, although the defects in the L1 HZO layer, as shown in Fig. 2j-l in the Manuscript, are expected to introduce shear strain in adjacent film layers, the $O_{//}$ - M_{010} structural transition actually occurs in the L2 HZO layer, rather than in the ZO layer between L1 and L2 layers. This observation further demonstrates that the interfacial effects play a crucial role in regulating the phase transition behaviors.

It is worthwhile to note that the conclusions discussed above is applicable to the epitaxial, single-crystalline $(\text{HZO-ZO})_6$ superlattice films in our Manuscript but the polycrystalline films. The high-density interfaces and defects in the as-grown polycrystalline films would introduce complex strain states and interfacial effects,

thereby influencing the intrinsic structural mechanisms of phase transitions among O -phase, T -phase and M -phase.

Changes in the revised Manuscript and revised Supplementary Materials:

To illuminate the roles of strain and interfacial effects, we have added some discussions in Lines 12-29 in Page 10 in the revised Manuscript. In addition, we have added the Fig. R1.6-R1.9 as Supplementary Fig. 12-15 in the revised Supplementary Materials.

Lines 12-29 in Page 10: “In Fig. 3, the O_{\perp} -phase exhibits out-of-plane polarization and undergoes reversible transitions with the T -phase. Conversely, in Fig. 2a-f, the $O_{//}$ -phase possesses in-plane polarization and progressively transforms into the M_{001} phase under irradiation. The discrepancy of the phase transition behaviors attributes to the different strain states and interfacial effects within the O_{\perp} -phase and $O_{//}$ -phase in the $(\text{HZO-ZO})_6$ superlattice films. The O_{\perp} -phase distributes throughout both the HZO and ZO layers, exhibiting an in-plane normal strain modulation with an alternating ‘contraction-expansion’ character, and is unaffected by interfacial effects (Supplementary Fig. 12b). In contrast, the $O_{//}$ -phase is constrained by the interface, primarily resides within the HZO layer, displays an out-of-plane normal strain modulation with an alternating ‘contraction-expansion’ character, and is accompanied by a significant out-of-plane lattice rotation (Supplementary Figs. 12c and 12e). The different strain states for the two types of O -phases are further confirmed by the quantitative strain analysis in Supplementary Fig. 13. Given the similarities and differences in atomic arrangements among the different phases, including the O -phase, T -phase, M_{001} -phase and M_{010} -phase (Supplementary Fig. 14-15), strain states and interfacial effects preferentially select the phase transformation pathways with higher structural compatibility and lower lattice distortion energy. Thereby, the O_{\perp} -phase tends to undergo reversible transitions with the T -phase (Fig. 3), whereas the $O_{//}$ -phase favors transitions with the M -phase (Fig. 2). However, phase transition behaviors in the polycrystalline films are complicated by the complex interplay of high-density interfaces and defects. Notably, O -phases with different polarization orientations themselves interconvert under electron beam irradiation, as demonstrated in Supplementary Fig. 16.”

Question and comment (R1.6): Does the presence shear strain leading to O - to M_{010} transformation mean that strain driven transformations are most likely to be O - to M -

phase transformations?

Reply to Question and comment (R1.6):

We appreciate the good questions from the referee. As noticed by the referee, the phase transition from *O*-phase to *M*₀₁₀-phase was realized by introducing the localized strain fields accompanied with the electron-beam-induced local defects. It attributes to the structural differences between *O*-phase and *M*₀₁₀-phase. As shown in Fig. R1.9a, the *O*//-phase features a 'one-wide-one-narrow' square arrangement of Hf/Zr sublattices and significant oxygen-ionic displacements. By contrast, the *M*₀₁₀-phase (Fig. R1.9c), viewed along the [010] zone axis, shows a distinct monoclinic angle and a unit cell composed of both parallelogram and square Hf/Zr sublattices. The transition from the *O*//-phase to the *M*₀₁₀-phase requires the stretching of the parallelogram sublattices into square grids, a process driven by shear strain. This confirms that shear strain facilitates the *O*//-phase to *M*₀₁₀-phase transition, in agreement with previous reports on its critical role¹¹.

However, the phase transition between *O*-phase and other phase follows a different mechanism, particularly the *T*-phase. As shown in Fig. R1.8a, the unit cell of *T*-phase features a cubic arrangement of Hf/Zr sublattices without oxygen ion displacement. In contrast, the *O*-phase exhibits a characteristic 'wide-narrow' alternating pattern in its square Hf/Zr sublattices, accompanied by significant oxygen ion displacement within the wider sublattices (Fig. R1.8b). Given the symmetry differences and lattice mismatch between the two phases^{11,14}, normal strain is anticipated to play a critical role in mediating the *O*-*T* structural transition. Therefore, the distinct symmetry and lattice parameters of these phases suggest that their transitions are governed by different strain states.

Question and comment (R1.7): The authors state that by understanding mechanisms of phase transformations it would allow for the structural design of HfO₂-based materials. A small discussion around how this work is applicable to devices and route forward for structural design would add a lot to the value of this paper.

Reply to Question and comment (R1.7):

We appreciate the constructive suggestion from the referee. As noticed by the referee, our work systematically clarifies the core mechanisms of structural phase transitions in HfO₂-based films.

The in-depth and systematic elucidation of the sublattice distortion pathways in the HfO₂-based superlattice films provides a fundamental framework for domain

engineering, enabling high-throughput fabrication with minimal performance variation. This insight opens a route toward tailoring ferroelectric domain configurations at the chip scale through precisely designed electrode geometries or localized strain fields, which may accelerate the development of neuromorphic synaptic components and dynamically reconfigurable radio-frequency circuits.

At the atomic scale, it is revealed that the polarization switching within the *O*-phase requires only dipole-order reversal of the oxygen sublattice, without the need for large-scale lattice reconstruction. This mechanism demonstrates the inherent potential of HfO₂-based ferroelectric devices for achieving ultra-low switching energy, providing a theoretical basis for the design of low-power memory cells.

Furthermore, the transition path from the *O*-phase to the *M*-phase is found to be highly dependent on the initial crystal orientation and projection direction. Given that the *M*-phase acts as a non-polar parasitic phase responsible for device performance degradation, strategic orientation control during growth can suppress its formation or restrict it to confined regions, thereby extending cyclability and improving retention in fabricated devices.

Beyond orientation effects, each phase transition pathway responds selectively to applied strain. By engineering interfacial stress and lattice mismatch in heterostructures, specific functional phases, such as the highly polar orthorhombic variant, can be stabilized at the nanoscale. This level of control further allows the periodic arrangement of ferroelectric and antiferroelectric domains, facilitating the creation of artificial superlattices with tailored electronic characteristics.

Changes in the revised Manuscript:

As suggested by the referee, we have added some discussions around how this work is applicable to devices and route forward for structural design in Lines 16-30, Page 13 and Lines 1-10, Page 14 in the revised Manuscript.

Lines 16-30, Page 13 and Lines 1-10, Page 14: “As a result, this mechanistic insight into structural phase transitions provides a crucial experimental foundation for both device applications and structural design in HfO₂-based thin films. Firstly, it is revealed that the polarization switching in the *O*-phase requires only dipole-order reversal of the oxygen sublattice, without the need for large-scale lattice reconstruction. This mechanism demonstrates the inherent potential of HfO₂-based ferroelectric

devices for achieving ultra-low switching energy, providing a theoretical basis for the design of low-power memory cells. Moreover, compared to mechanisms involving substantial cation displacements, the low-energy mechanism is likely more robust against thermal fluctuations, which may enhance device endurance at elevated temperatures. Secondly, the transition path from the *O*-phase to the *M*-phase is found to be highly dependent on the initial crystal orientation and projection direction. Given that the *M*-phase acts as a non-polar parasitic phase responsible for device performance degradation, strategic orientation control during growth can suppress its formation or restrict it to confined regions, even at high temperatures, thereby improving cycling endurance and retention in fabricated devices. Thirdly, the observed *FE-AFE* reversibility and *O*-phase instability towards non-polar phases provide a microscopic basis for understanding device phenomena such as wake-up, fatigue, and endurance degradation. Therefore, these insights offer critical guidance for designing more reliable hafnia-based ferroelectric memories. Finally, it is found that the phase transition pathways respond selectively to strain states. By engineering interfacial stress and lattice mismatch in heterostructures, desired functional phases, such as the highly polar orthorhombic variant, can be stabilized at the nanoscale, effectively countering thermally induced phase instability. This level of control further enables the periodic arrangement of ferroelectric and antiferroelectric domains, paving the way for creating artificial superlattices with customized electronic properties.”

Question and comment (R1.8): It seems as though these time-dependent changes are very dynamic. What are the implications for device operation at elevated temperatures.

Reply to Question and comment (R1.8):

We appreciate the insightful question from the referee. We fully agree with the referee that the dynamic nature of the phase transitions and their implications for high-temperature device operation are the critical consideration for the practical application of HfO₂-based ferroelectric devices.

At elevated temperatures, enhanced ionic mobility and accelerated kinetics are expected to intensify the time-dependent structural evolutions. This could potentially

lead to some adverse effects on the device applications. For example, the high temperature would induce the uncontrolled phase transitions from the polar *O*-phase to non-polar phase, causing a loss of polarization. Meanwhile, the resulting phase instability may accelerate fatigue and device failure during repeated switching cycles. Besides, the stability of a programmed polarization state could be compromised over time at high temperatures, leading to the data retention issues.

Crucially, our work provides fundamental insights and novel design principles to alleviate these challenges. The nanoscale structural mechanisms elucidated in the Manuscript offer useful guidance for enhancing thermal stability. As revealed in our Manuscript, the phase transition behaviors among *O*-phase, *M*-phase and *T*-phase depend on the initial *O*-phase direction and projection direction of *M*-phase, as shown in Fig. 2-3 in the Manuscript. Thus, by precisely engineering the crystal orientation, the phase evolution could be designed to suppress the formation of the detrimental *M*-phase, even at elevated temperatures. Besides, the polarization switching within the *O*-phase is confirmed to require only a localized oxygen sub-lattice dipole-order reversal. This low-energy mechanism is less disruptive to the lattice and may be more robust against thermal agitation compared to mechanisms involving large-scale cation movements, potentially leading to better endurance at high temperatures. Furthermore, the asynchronous sublattice distortion mechanism illuminated in our Manuscript is expected to provide guidance for targeted material design. For instance, strain engineering in superlattice films or interface design can be used to selectively pin specific sublattices, thereby stabilizing the desired ferroelectric phase against thermal-driven transitions.

In summary, while the dynamic phase evolution presents a challenge for high-temperature operation, our findings transform this challenge from an uncontrollable phenomenon into a manageable design parameter. The mechanistic understanding we provide lays the groundwork for designing HfO₂-based films with superior thermal stability.

Changes in the revised Manuscript:

To illuminate the implications for device operation at elevated temperatures, we have added some discussions in Lines 16-30, Page 13 and Lines 1-10, Page 14 in the revised Manuscript.

Lines 16-30, Page 13 and Lines 1-10, Page 14: “**As a result**, this mechanistic **insight into** structural phase transitions provides a crucial experimental **foundation** for

both device applications and structural design in HfO₂-based thin films. Firstly, it is revealed that the polarization switching in the *O*-phase requires only dipole-order reversal of the oxygen sublattice, without the need for large-scale lattice reconstruction. This mechanism demonstrates the inherent potential of HfO₂-based ferroelectric devices for achieving ultra-low switching energy, providing a theoretical basis for the design of low-power memory cells. Moreover, compared to mechanisms involving substantial cation displacements, the low-energy mechanism is likely more robust against thermal fluctuations, which may enhance device endurance at elevated temperatures. Secondly, the transition path from the *O*-phase to the *M*-phase is found to be highly dependent on the initial crystal orientation and projection direction. Given that the *M*-phase acts as a non-polar parasitic phase responsible for device performance degradation, strategic orientation control during growth can suppress its formation or restrict it to confined regions, even at high temperatures, thereby improving cycling endurance and retention in fabricated devices. Thirdly, the observed *FE-AFE* reversibility and *O*-phase instability towards non-polar phases provide a microscopic basis for understanding device phenomena such as wake-up, fatigue, and endurance degradation. Therefore, these insights offer critical guidance for designing more reliable hafnia-based ferroelectric memories. Finally, it is found that the phase transition pathways respond selectively to strain states. By engineering interfacial stress and lattice mismatch in heterostructures, desired functional phases, such as the highly polar orthorhombic variant, can be stabilized at the nanoscale, effectively countering thermally induced phase instability. This level of control further enables the periodic arrangement of ferroelectric and antiferroelectric domains, paving the way for creating artificial superlattices with customized electronic properties.”

Question and comment (R1.9): In addition, there are some minor improvements to language and grammar needed.

Reply to Question and comment (R1.9):

We sincerely thank the referee for this valuable comment and for pointing out the need for language polishing. We have carefully reviewed the entire Manuscript and

implemented comprehensive English language edits to address grammatical errors, improve sentence structures, and enhance overall clarity and fluency. These changes have been made throughout the text. We believe the Manuscript is now much improved and meets the language standards.

Reply to Reviewer #2:

We appreciate the positive comment by the referee that “The study is technically rigorous and supported by high-quality microscopy data” and the accurate summary of our work that “The authors report electron-beam-induced polymorphic phase transitions in HfO₂-based superlattices and introduce an asynchronous sublattice distortion mechanism. Transmission electron microscopy demonstrates that preferential sublattice distortions drive transitions among orthorhombic, tetragonal, and monoclinic phases. In addition, the polar-antipolar transition within the orthorhombic phase is attributed solely to the reversal of oxygen sublattice dipole order”.

In the meanwhile, the referee also raises several specific questions and comments which are summarized into five aspects. We fully understand the referee’s concerns, and here we have addressed all the questions and discussed all the comments one by one in the following. The revisions are written in **RED** in the revised manuscript and the revised Supplementary Materials.

Question and comment (R2.1): The electron-beam-induced phase transitions are intriguing, but ferroelectric devices are operated under applied electric fields, not electron irradiation. The work would gain much greater importance if the structural change mechanisms under voltage operation were investigated. What structural changes would the authors expect during electric-field experiments, and how might they differ from those observed under electron-beam excitation?

Reply to Question and comment (R2.1):

We appreciate the constructive suggestions from the referee. As noticed by the referee, we have provided the electron-beam-induced phase transitions of the hafnia-based superlattice films grown on YSZ (001) substrate in the Manuscript. We fully agree with the referee that the comparisons of phase transition behaviors under different stimuli including the electron-beam irradiation and external electric field are crucial.

However, the prerequisite for conducting *in-situ* electrical TEM/STEM experiments is that the sample itself is conductive, or that electrodes are fabricated on the surface of the insulating material to enable electrical conduction. In this study, we epitaxially grew single-crystal (HZO-ZO)₆ superlattice thin films on YSZ (001) substrates. Since both the YSZ substrate and the (HZO-ZO)₆ superlattice film are non-conductive, and no bottom electrode material was introduced between them, thus it is a pity that *in-situ* electrical TEM/STEM experiments could not be performed.

If a bottom electrode material (such as the commonly used perovskite-structured

LSMO or SrRuO₃) was epitaxially grown between the YSZ substrate and the superlattice film, its structural mismatch with the fluorite-type HfO₂-based film would lead to the growth of a polycrystalline and multiphase hafnia-based superlattice structure. Such structures contain complex interfaces and defects, which would significantly affect the structural evolution behaviors of the HfO₂-based films under an applied electric field, making the observations unlikely to reflect the intrinsic field-induced structural phase transition mechanism.

Despite the aforementioned challenges, previous *in-situ* electrical studies on polycrystalline and multiphase hafnia-based films have shown that the structural evolution induced by electric field stimulation exhibits certain similarities to the phase transition behavior induced by electron beam irradiation reported in our Manuscript. For instance, the *in-situ* HRTEM experiments conducted on a TiN/HZO (15 nm)/TiN capacitor structure revealed that the *T*-phase transformed into the *O*-phase under a positive electric field, and reverted to the *T*-phase after the electric field is removed. This reversible phase transition process is consistent with the *T-O* phase transition behaviors we observed under electron beam irradiation (Fig. 3 in our Manuscript). The difference lies in the use of the iDPC-STEM imaging mode for *in-situ* observation in our Manuscript, which not only confirmed the real-time phase transition process but also revealed the mechanism of asynchronous sublattice distortion during the phase transition. Furthermore, ex-situ STEM analysis performed by the same research group on the same TiN/HZO (15 nm)/TiN capacitor structure after electric field cycling showed that the “wake-up” process corresponds to an *AFE*→*FE* phase transition, while the “fatigue” process corresponds to an *FE*→*AFE* phase transition. These results are also consistent with the reversible *FE-AFE* phase transition behavior we reported under electron beam irradiation (Fig. 4 in our Manuscript).

The above consistency of phase transition behaviors suggests that, despite the different physical stimuli, the material system is driven into a similar free energy landscape, thereby following comparable phase transition pathways. Thus, whether it is the external electric field in devices or the built-in electric field induced by the electron beam irradiation, both ultimately act on the ions in the crystal lattice. They both could overcome energy barriers through electrostatic interactions, promoting ion displacement and ultimately achieving the phase transition.

Changes in the revised Manuscript:

We have added some discussions in Lines 11-20, Page 14 in the revised Manuscript.

Lines 11-20, Page 14: “It should be noted that this study focuses specifically on the mechanism of electron beam irradiation-induced structural phase transitions. While the timescales of electron beam irradiation-induced transitions are not directly comparable to the nanosecond-scale field-induced switching in devices, the atomistic pathways revealed here are expected to be fundamental. The asynchronous sublattice distortion mechanisms are governed by the intrinsic energy landscape of HfO₂-based materials, which also dictates the response under an electric field. Even though, we fully recognize that other types of stimuli may lead to differences in phase transition types and kinetic processes. Further research on this basis, particularly exploring the influence of *in-situ* electric fields under different application modes, is of significant importance.”

Question and comment (R2.2): While the authors propose a systematic framework for phase transitions, the unique contribution of the asynchronous sublattice distortion model compared to earlier experimental and theoretical studies is not sufficiently emphasized. What new insight does this model provide beyond existing reports?

Reply to Question and comment (R2.2):

We sincerely thank the referee for this insightful comment and for highlighting the importance of clarifying the unique contribution of our proposed asynchronous sublattice distortion model. As noticed by the referee, we proposed the asynchronous sublattice distortion mechanism underlying the complex phase transitions in HfO₂-based superlattice films, including the reversible transitions of *O-M*, *O-T* and polar *O*-antipolar *O*-phases. Our asynchronous sublattice distortion model provides several fundamentally new and unifying insights that go beyond earlier experimental and theoretical reports, which have largely focused on individual phase transitions or energy-based explanations without in-depth resolving the atomistic sequence of ionic motions for multiple phase transitions. Below, we will summarize the key advances offered by our model.

Firstly, for the structural phase transition between *O*-phase and *M*-phase, previous studies on ZrO₂ nanocrystals have revealed that electron-beam irradiation induces a *M*-like intermediate state (*M'*), and the transition between the *O*- and *M'*- phases is reversible¹¹. However, that work primarily focused on lattice shear behavior and strain evolution, namely the evolution pathway of the cations (Zr), without clarifying the

behavior of the anions (oxygen ions) or their distortion sequence during the phase transition. The asynchronous sublattice distortion model proposed in our Manuscript clearly demonstrates that the lattice distortions of the cations and anions during the O - M phase transition are not synchronous but follow a defined sequence. More importantly, we reveal that the O - M phase transition pathway is not unique. Different projection directions of the M -phase lead to distinct transition mechanisms and priorities in sublattice distortion. We further emphasize the critical role of shear strain in inducing different types of O - M phase transitions, as shown in Fig. 2 in the Manuscript. Notably, the reversible transition between the M_{010} - and O - phases exhibits a markedly asymmetric sublattice distortion sequence, which stands in sharp contrast to the symmetric and reversible phase transition behavior observed between M_{001} - and O -phases. Therefore, our study provides a systematic and in-depth understanding of the structural mechanisms underlying O - M phase transitions.

Secondly, regarding the structural transformation between the O -phase and T -phase, although several theoretical and experimental studies have been conducted, significant discrepancies exist among the reported results. Theoretically, one first-principles study indicated that in HfO_2 , the transition from the T -phase to the polar O -phase is kinetically more favorable than the transformation to the thermodynamically more stable antipolar O -phase¹⁵. Another study, based on lattice dynamics and order parameter coupling, proposed a transition pathway from T -phase to antipolar O -phase and finally to the existence of polar O -phase and antipolar O -phase¹⁶. To resolve these theoretical inconsistencies, experimental results have provided valuable explanations. Research on Lu-doped $\text{Hf}_{0.6}\text{Zr}_{0.4}\text{O}_2$ bulk single crystals suggested that the annealing temperature can modulate the transition sequence of the O -phase into either polar or antipolar O -phases. Through lattice analysis, it was inferred that during the T - O transformation, oxygen ions displace first, followed by the distortion of the cation sublattice¹⁷. In contrast, another study on ZrO_2 nanocrystals reported that the T - O transformation is initiated by rearrangement of the Zr cation sublattice, prior to oxygen ion displacement¹². These experimental discrepancies may stem from the differences in material systems and phase distribution morphology, as well as methodological variations. The former study on doped $\text{Hf}_{0.6}\text{Zr}_{0.4}\text{O}_2$ bulk single crystals inferred the transition pathway from regional structural comparisons, which may not reflect the real-time evolution process. Although the latter study on ZrO_2 nanocrystals tracked the same location, the initial T -phase was confined to a single unit cell and surrounded by the O -phase. Furthermore, the strong surface/interface effects inherent to nanocrystals could perturb the transition pathway and preclude the observation of reversibility. In comparison, by focusing on the Hf-based superlattice single-crystal films, we effectively minimized interference from complex interfaces and defects in nanocrystals,

enabling the real-time observation of the entire reversible T - O phase transition process under electron-beam irradiation, as shown in Fig. 3 in the Manuscript. We propose that the reversible T - O phase transition follows an asynchronous mechanism dominated by preferential distortion of the Hf/Zr cation sublattice. Furthermore, we revealed that the T - O phase transition tends to preferentially form the polar O -phase, whereas during the O - T phase transition, the antipolar O -phase first converts to the polar O -phase before transitioning to the T -phase. Thus, our experiments clearly demonstrate the selectivity between polar O -phase and antipolar O -phase pathways in the reversible T - O phase transition.

Thirdly, regarding the structural evolution between the polar O -phase and antipolar O -phase, Our study in the present Manuscript confirms that the transition between different variants of the O -phase involves only oxygen ion displacement to achieve unit-cell-scale polarization switching and reversible polar-antipolar transition, which is consistent with existing reports¹². However, by comparing different irradiation processes, we further found that the polar-antipolar transition occurs more readily, thereby deepening the understanding of its phase transition kinetics.

While the preceding discussion has focused on the electron-beam-induced phase transitions to highlight the novelty of the asynchronous sublattice distortion model proposed in our Manuscript, significant controversy also exists in interpreting the structure-property relationship of HfO₂-based materials under electric-field stimulation. Various mechanisms have been proposed for the wake-up and fatigue phenomena in HfO₂-based capacitors. For example, One study attributes wake-up to defect redistribution and fatigue to domain pinning and new defect generation¹⁸. Another study links wake-up to phase structural changes in the bulk or at interfaces and fatigue to defect accumulation and domain pinning¹⁹. Yet in-depth structural study suggests that both wake-up and fatigue in Hf_{0.5}Zr_{0.5}O₂ capacitors result from reversible antipolar-polar O -phase transitions¹³. These controversies likely originate from the complex defect and interface structures in capacitors, which complicate the accurate assessment of field-driven structural evolution. It is expected that the relationship of different phase transitions with wake-up/fatigue behaviors in various studies may be influenced by the orientation distribution of the O -phase. Our study reveals that the polarization direction of the O -phase dictates its transition pathway under an external field. For instance, the $O_{//}$ -phase tends to undergo a reversible transformation with the M -phase, whereas the O_{\perp} -phase preferentially switches reversibly with the T -phase. This understanding provides a new structural perspective for understanding the mechanistic discrepancies reported in prior studies.

In summary, by proposing the asynchronous sublattice distortion model, this work

integrates the diverse and complex structural phase transition behaviors in HfO₂-based materials into a unified theoretical framework. It systematically elucidates numerous controversies in prior researches, and provides a deeper, atomistic understanding of phase transition mechanisms. Our research moves beyond phenomenological description by establishing a predictive, ionically-resolved model that not only identifies the occurrence of phase transitions but also reveals their dynamic processes at the atomic scale. This marks a crucial step toward the rational design of phase-stable HfO₂-based ferroelectric materials.

Changes in the revised Manuscript:

To illuminate the new insight of the asynchronous sublattice distortion model proposed in our Manuscript compared to earlier experimental and theoretical studies, we have added some discussions in the revised Manuscript.

Lines 4-17, Page 3: “Structural linkages between ferroelastic switching and the *O*-to-*M* transition have been elucidated in ZrO₂ nanocrystals¹⁴. However, existing researches on the *O*-*M* phase transition has predominantly focused on the lattice shear behavior of the cation sublattice and the corresponding strain evolution, while overlooking the role of the anions (oxygen ions), thus leaving the underlying structural mechanism incompletely understood. Furthermore, the specific sequence of cation and anion sublattice distortions during the *T*-*O* phase transition remains a subject of debate, as experimental findings across different HfO₂-based systems are inconsistent^{13,23}. Beyond these fundamental transitions, the structural origins of wake-up and fatigue under electric fields, though frequently attributed to phase transitions, are also controversial. For instance, structural studies on Hf_{0.5}Zr_{0.5}O₂ capacitors attribute these phenomena to reversible transitions between ferroelectric (FE) and antiferroelectric (AFE) *O*-phases²⁴. In contrast, investigations on HfO₂ capacitors link wake-up to changes in the phase structure (involving *M*-, *O*-, and *T*-phases) in the bulk and at interfaces, while ascribing fatigue to defect generation and domain-wall pinning²⁵.”

Lines 11-15, Page 13: “In a word, the reported rich phase transitions among *O*-phase and other non-polar phases under electron beam irradiation confirm the low energy barriers among these phases as reported previously³⁵, and further reveal the

asynchronous sub-lattice distortion model governing these pathways. This model integrates the diverse and complex structural phase transition behaviors in HfO₂-based materials into a unified theoretical framework, systematically elucidating numerous controversies in prior researches and providing a deeper, atomistic understanding of phase transition mechanisms.”

Question and comment (R2.3): The rationale for using a superlattice structure is not entirely clear. Why are ZrO₂ layers included, and what is their functional role in the observed transitions? How would the results be expected to differ in a single HZO layer film without superlattice modulation?

Reply to Question and comment (R2.3):

We appreciate the constructive discussions and good questions from the referee. As noticed by the referee, we illuminated the detailed phase transition behaviors of the (HZO-ZO)₆ epitaxial superlattice thin film under electron-beam irradiation. In hafnia-based materials, the ferroelectricity primarily originates from the metastable *O*-phase (*Pca2₁*). Therefore, stabilizing this polar *O*-phase and understanding its phase transitions to non-polar phases are critical. This fundamental understanding is essential for enhancing the macroscopic ferroelectric properties of hafnia-based materials and advancing their commercial applications. To eliminate the interference from high-density defects and interfaces in polycrystalline films to the analysis of phase transition mechanisms, it is necessary to construct a simpler single-crystal film system. Thus, we chose to epitaxially grow hafnia-based films on YSZ (001) substrates. YSZ exhibits similar structural symmetry and a high degree of lattice matching with HZO, ZrO₂, and HfO₂. This compatibility enables the growth of high-quality epitaxial single-crystal films, thereby avoiding interference from the complex microstructure of polycrystalline films in the analysis of phase transition mechanisms.

On the other hand, to elucidate the structural phase transition mechanisms between the *O*-phase and other non-polar phases in hafnia-based films, it is first necessary to construct a film system with the metastable *O*-phase being stabilized. Although a single-layer HZO film has a simpler structure, preliminary studies have shown that single-layer HZO single-crystal films epitaxially grown on YSZ (001) substrates are predominantly composed of the non-polar *M*-phase (*P2₁/c*), making it difficult to stabilize the metastable orthorhombic ferroelectric phase in such films (Fig. R2.1). Superlattice thin-film systems, as typical size-confined systems, are constructed by periodically alternating the layer growth of different materials, introducing significant

chemical, strain, symmetry, charge, and polarization discontinuities between layers²⁻⁴. These result in strong interfacial and interlayer interactions. Consequently, they hold the potential to overcome the physical limitations of bulk materials, stabilize new structures, and induce novel physical properties comparing with the pure HZO films. Thereby, we constructed a (HZO-ZO)₆ superlattice single-crystal film system, successfully stabilized the polar *O*-phase, and further systematically investigated the phase transition behaviors and structural mechanisms between the *O*-phase and other phases using electron-beam irradiation experiments in an aberration-corrected transmission electron microscope.

Fig. R2.1| Atomic-resolved HAADF-STEM image of the 10 nm Hf_{0.5}Zr_{0.5}O₂ film, with the nonpolar monoclinic phase (*P2₁/c*) being stabilized.

In the present Manuscript, it reveals a pronounced spatial selectivity in the phase distribution and transition behaviors within the superlattice films. Specifically, the *O*_{//}-phase preferentially stabilizes in the HZO layers, whereas the *O*_⊥-phase and the *T*-phase can extend across the interfaces, coexisting in both HZO and ZO layers (Fig. 1h).

Under electron beam irradiation, the structural transitions between different phases further highlight the regulatory role of interfacial effects. The transformation of the *O*_{//} phase to the *M*-phase is confined by the HZO/ZO interfaces, resulting in the formation of the *M*-phase exclusively within the HZO layers (Fig. 2). In contrast, the transition from the *O*_⊥-phase to the *T*-phase is not restricted by the interfaces. This phase transformation can propagate across the HZO/ZO interfaces, occurring simultaneously in both layers (Fig. 3).

Especially, although the *O*_{//} phase predominantly resides in the HZO layers, the *AFE-O* phase can be induced and stabilized within the ZO layers after sufficient electron beam irradiation (Figs. 4i-j). These findings demonstrate that the introduction

of ZO layers plays a critical role not only in stabilizing the *O*-phase but also in governing the spatial evolution of phase transition pathways within the superlattice films.

Changes in the revised Manuscript:

To illuminate the reasons for using a superlattice structure, we have added some discussions in Lines 16-21, Page 4 in the revised Manuscript.

Lines 16-21, Page 4: “Given that $\text{Y}_2\text{O}_3\text{:ZrO}_2$ (YSZ) substrates share the fluorite structure with HfO_2 - and ZrO_2 -based materials, the epitaxial growth of HfO_2 -based thin films on YSZ substrates enables superior crystalline quality. **Furthermore, superlattice thin-film systems, as typical size-confined systems, are constructed by periodically alternating the layer growth of different materials, displaying the strong interfacial and interlayer interactions. They are expected to overcome the physical limitations of bulk materials, for example, stabilizing the desired polar structures and inducing novel physical properties in HfO_2 -based materials^{27,28}.**”

Question and comment (R2.4): The manuscript reports facile phase transitions among the different polymorphs. Could the authors provide a discussion of the theoretical or experimental energy differences between these phases?

Reply to Question and comment (R2.4):

We appreciate the constructive suggestion from the referee. As noticed by the referee, we have discussed multiple reversible phase transitions among the different polymorphs in HfO_2 -based superlattice films, including the *O*- M_{001} , *O*- M_{010} , *O*-*T* and *FE*-*AFE* transitions.

Especially, the *O*- M_{001} and *O*- M_{010} transformation processes can be uniformly categorized as the structural transitions between the *O*-phase and *M*-phase. In Fig. 2, the initial structure of the superlattice thin film presents the nonpolar cubic lattice, which may correspond to either the *C*-phase or *T*-phase. Since the theoretical energy of the *C*-phase is much higher than that of *T*-phase^{12,20}, the initial structure of the film is considered to be *T*-phase. It is revealed that under electron beam irradiation, the *T*-phase tends to first transform into the *O*-phase, and then further into the *M*-phase. This transition pathway is consistent with previous reports of theoretical calculations. It was revealed that, under fixed lattice parameters, the energy required for the *T*-*O* transformation was much lower than that for a direct transition from *T*-phase to *M*-

phase²¹.

Regarding the phase transition between the *T*-phase and *O*-phase as shown in Fig. 3 in the Manuscript, it has been found that the *T*-phase preferentially transforms into the *FE-O* phase. In the reverse transformation, the *AFE-O* phase first transforms into the *FE-O* phase before transitioning to the *T*-phase. This behavior is attributed to the fact that the energy barrier for the transition between *T*-phase and *FE-O* phase is lower than that for the transition between *T*-phase and *AFE-O* phase^{15,21}.

With respect to the structural transition between different variants of the *O*-phase, namely *FE-O* and *AFE-O*, it is observed that under the same electron beam irradiation duration, the area of the *FE-AFE* transition is significantly larger than that of the *AFE-FE* transition (Fig. 4 in the Manuscript), indicating that the *AFE-O* phase occupies a lower energy state. This finding is consistent with previous theoretical calculations¹¹.

Changes in the revised Manuscript:

To explain the observed phase transition behaviors from the energetic perspective, we have added some discussions in the revised Manuscript.

Lines 15-19, Page 6: “The dominant structure in HZO layers in the as-grown superlattice film (Fig. 2a) presents the nonpolar cubic lattice, which may correspond to either the *C*-phase or *T*-phase. Since the theoretical energy of the *C*-phase is much higher than that of *T*-phase³², the initial structure of the film is considered to be *T*-phase, which is also consistent with the XRD result in Fig. 1g. Figure 2a-f tracks the phase evolution among *T*-, *O*//-, and *M*-phases in the HZO layer under electron beam irradiation.”

Lines 24-30, Page 6 and Lines 1-3, Page 7: “The predominant pathway is that *T*-phase progressively transforms into *O*//-phase (pink mask) with increasing irradiation time, followed by the emergence of an intermediate state M_{001}' (green mask), and ultimately the formation of the *M*-phase projected along [001] direction (M_{001} , cyan mask). This transition pathway is consistent with previous reports of density functional theory calculations. It was revealed that the energy required for the *T-O* transformation was much lower than that for a direct transition from *T*-phase to *M*-phase under the fixed lattice parameters of HfO_2 ³³. Crucially, the transformation from *O*-phase to M_{001} -phase proceeds through an intermediary phase, with the *O*-phase first evolving into the

M₀₀₁' phase before converting to the final *M₀₀₁* phase. The *M₀₀₁'* phase is distinct from *M₀₀₁* in its lattice distortion, as evidenced in Supplementary Fig. 10. Understanding the sequence of cationic (Hf/Zr) and anionic (O) sublattice distortions during this process is key to elucidating the underlying mechanism.”

Line 10-12, Page 11: “When transforming from the *T*-phase to the *O_⊥*-phase, the *T*-phase preferentially transforms into the *FE-O_⊥* phase rather than *AFE-O_⊥* phase (yellow rectangle in Supplementary Fig. 11b), which is supported by previous theoretical predictions that energy barrier for the transition between *T*-phase and *FE-O* phase is lower than that for the transition between *T*-phase and *AFE-O* phase^{33,34}.”

Lines 2-8, Page 12: “More critically, under identical irradiation durations, the phase-transition volume is larger for the *FE*-to-*AFE* transition (Figs. 4e-f) than for the reverse *AFE*-to-*FE* process (Figs. 4g-h). This asymmetry provides direct experimental evidence for the lower energy state and greater stability of the *AFE-O* phase, thereby validating earlier theoretical predictions^{14,34}. Furthermore, Fig. 1h shows that the *O*-phase is laterally confined within the HZO layers, whereas the adjacent ZO layers maintain the *T*-phase structure.”

Question and comment (R2.5): The manuscript frequently uses the phrase "It is noteworthy that" which appears repetitive. The authors should revise such instances to improve readability and stylistic variety.

Reply to Question and comment (R2.5):

We appreciate the careful reading and constructive suggestions from the referee. In the revised Manuscript, we have updated these statements to improve the readability and stylistic variety of the Manuscript.

Reply to Reviewer #3:

We appreciate the positive comment by the referee that “Asynchronous sublattice distortion mechanisms seemed to be interesting. This proposal is a timely topic in hafnia ferroelectricity and worthwhile to be reported” and “In my opinion, this study demonstrated clear experimental results to be suited for Nature Communications and might be publishable”.

In the meanwhile, the referee also raises several specific questions and comments which are summarized into four aspects. We fully understand the referee’s concerns, and here we have addressed all the questions and discussed all the comments one by one in the following. The revisions are written in **RED** in the revised manuscript and the revised Supplementary Materials.

Question and comment (R3.1): The proposed mechanism of asynchronous sublattice distortion, distinguishing Hf/Zr-initiated versus O-initiated phase transitions, is both novel and compelling. This conceptual framework can have broad implications beyond HfO₂-based materials, possibly informing studies on ZrO₂, TiO₂, and other oxide-based functional materials. The use of epitaxial superlattices as a model system is particularly commendable, as it effectively isolates the intrinsic transformation behavior from extrinsic effects such as grain boundaries and interfacial disorder. For the general readers, the discussion could benefit from a clearer comparison to previously proposed phase transition models, particularly those involving field-induced switching and thermodynamic simulations.

Reply to Question and comment (R3.1):

We sincerely thank the referee for this insightful comment and constructive suggestion. As noticed by the referee, we proposed the asynchronous sublattice distortion mechanism underlying the complex phase transitions in HfO₂-based superlattice films, including the reversible transitions of *O-M*, *O-T* and polar *O*-antipolar *O*-phases. Our asynchronous sublattice distortion model provides several fundamentally new and unifying insights that go beyond previously proposed phase transition models, which have largely focused on individual phase transitions or energy-based explanations without in-depth resolving the atomistic sequence of ionic motions for multiple phase transitions. Below, we will summarize the key advances offered by our model.

Firstly, for the structural phase transition between *O*-phase and *M*-phase, previous studies on ZrO₂ nanocrystals have revealed that electron-beam irradiation induces a *M*-

like intermediate state (M'), and the transition between the O - and M' - phases is reversible¹¹. However, that work primarily focused on lattice shear behavior and strain evolution, namely the evolution pathway of the cations (Zr), without clarifying the behavior of the anions (oxygen ions) or their distortion sequence during the phase transition. The asynchronous sublattice distortion model proposed in our Manuscript clearly demonstrates that the lattice distortions of the cations and anions during the O - M phase transition are not synchronous but follow a defined sequence. More importantly, we reveal that the O - M phase transition pathway is not unique. Different projection directions of the M -phase lead to distinct transition mechanisms and priorities in sublattice distortion. We further emphasize the critical role of shear strain in inducing different types of O - M phase transitions, as shown in Fig. 2 in the Manuscript. Notably, the reversible transition between the M_{010} - and O - phases exhibits a markedly asymmetric sublattice distortion sequence, which stands in sharp contrast to the symmetric and reversible phase transition behavior observed between M_{001} - and O -phases. Therefore, our study provides a systematic and in-depth understanding of the structural mechanisms underlying O - M phase transitions.

Secondly, regarding the structural transformation between the O -phase and T -phase, although several theoretical and experimental studies have been conducted, significant discrepancies exist among the reported results. Theoretically, one first-principles study indicated that in HfO_2 , the transition from the T -phase to the polar O -phase is kinetically more favorable than the transformation to the thermodynamically more stable antipolar O -phase¹⁵. Another study, based on lattice dynamics and order parameter coupling, proposed a transition pathway from T -phase to antipolar O -phase and finally to the existence of polar O -phase and antipolar O -phase¹⁶. To resolve these theoretical inconsistencies, experimental results have provided valuable explanations. Research on Lu-doped $\text{Hf}_{0.6}\text{Zr}_{0.4}\text{O}_2$ bulk single crystals suggested that the annealing temperature can modulate the transition sequence of the O -phase into either polar or antipolar O -phases. Through lattice analysis, it was inferred that during the T - O transformation, oxygen ions displace first, followed by the distortion of the cation sublattice¹⁷. In contrast, another study on ZrO_2 nanocrystals reported that the T - O transformation is initiated by rearrangement of the Zr cation sublattice, prior to oxygen ion displacement¹². These experimental discrepancies may stem from the differences in material systems and phase distribution morphology, as well as methodological variations. The former study on doped $\text{Hf}_{0.6}\text{Zr}_{0.4}\text{O}_2$ bulk single crystals inferred the transition pathway from regional structural comparisons, which may not reflect the real-time evolution process. Although the latter study on ZrO_2 nanocrystals tracked the same location, the initial T -phase was confined to a single unit cell and surrounded by the O -phase. Furthermore, the strong surface/interface effects inherent to nanocrystals could

perturb the transition pathway and preclude the observation of reversibility. In comparison, by focusing on the HfO₂-based superlattice single-crystal films, we effectively minimized interference from complex interfaces and defects in nanocrystals, enabling the real-time observation of the entire reversible *T-O* phase transition process under electron-beam irradiation, as shown in Fig. 3 in the Manuscript. We propose that the reversible *T-O* phase transition follows an asynchronous mechanism dominated by preferential distortion of the Hf/Zr cation sublattice. Furthermore, we revealed that the *T-O* phase transition tends to preferentially form the polar *O*-phase, whereas during the *O-T* phase transition, the antipolar *O*-phase first converts to the polar *O*-phase before transitioning to the *T*-phase. Thus, our experiments clearly demonstrate the selectivity between polar *O*-phase and antipolar *O*-phase pathways in the reversible *T-O* phase transition.

Thirdly, regarding the structural evolution between the polar *O*-phase and antipolar *O*-phase, Our study in the present Manuscript confirms that the transition between different variants of the *O*-phase involves only oxygen ion displacement to achieve unit-cell-scale polarization switching and reversible polar-antipolar transition, which is consistent with existing reports¹². However, by comparing different irradiation processes, we further found that the polar-antipolar transition occurs more readily, thereby deepening the understanding of its phase transition kinetics.

While the preceding discussion has focused on the electron-beam-induced phase transitions to highlight the novelty of the asynchronous sublattice distortion model proposed in our Manuscript, significant controversy also exists in interpreting the structure-property relationship of HfO₂-based materials under electric-field stimulation. Various mechanisms have been proposed for the wake-up and fatigue phenomena in HfO₂-based capacitors. For example, One study attributes wake-up to defect redistribution and fatigue to domain pinning and new defect generation¹⁸. Another study links wake-up to phase structural changes in the bulk or at interfaces and fatigue to defect accumulation and domain pinning¹⁹. Yet in-depth structural study suggests that both wake-up and fatigue in Hf_{0.5}Zr_{0.5}O₂ capacitors result from reversible antipolar-polar *O*-phase transitions¹³. These controversies likely originate from the complex defect and interface structures in capacitors, which complicate the accurate assessment of field-driven structural evolution. It is expected that the relationship of different phase transitions with wake-up/fatigue behaviors in various studies may be influenced by the orientation distribution of the *O*-phase. Our study reveals that the polarization direction of the *O*-phase dictates its transition pathway under an external field. For instance, the *O*_{//}-phase tends to undergo a reversible transformation with the *M*-phase, whereas the *O*_⊥-phase preferentially switches reversibly with the *T*-phase. This understanding

provides a new structural perspective for understanding the mechanistic discrepancies reported in prior studies.

In summary, by proposing the asynchronous sublattice distortion model, this work integrates the diverse and complex structural phase transition behaviors in HfO₂-based materials into a unified theoretical framework. It systematically elucidates numerous controversies in prior researches, and provides a deeper, atomistic understanding of phase transition mechanisms. Our research moves beyond phenomenological description by establishing a predictive, ionically-resolved model that not only identifies the occurrence of phase transitions but also reveals their dynamic processes at the atomic scale. This marks a crucial step toward the rational design of phase-stable HfO₂-based ferroelectric materials.

Changes in the revised Manuscript:

To illuminate the new insight of the asynchronous sublattice distortion model proposed in our Manuscript compared to previously proposed phase transition models, we have added some discussions in the revised Manuscript.

Lines 4-17, Page 3: “Structural linkages between ferroelastic switching and the *O*-to-*M* transition have been elucidated in ZrO₂ nanocrystals¹⁴. However, existing researches on the *O*-*M* phase transition has predominantly focused on the lattice shear behavior of the cation sublattice and the corresponding strain evolution, while overlooking the role of the anions (oxygen ions), thus leaving the underlying structural mechanism incompletely understood. Furthermore, the specific sequence of cation and anion sublattice distortions during the *T*-*O* phase transition remains a subject of debate, as experimental findings across different HfO₂-based systems are inconsistent^{13,23}. Beyond these fundamental transitions, the structural origins of wake-up and fatigue under electric fields, though frequently attributed to phase transitions, are also contentious. For instance, structural studies on Hf_{0.5}Zr_{0.5}O₂ capacitors attribute these phenomena to reversible transitions between ferroelectric (FE) and antiferroelectric (AFE) *O*-phases²⁴. In contrast, investigations on HfO₂ capacitors link wake-up to changes in the phase structure (involving *M*-, *O*-, and *T*-phases) in the bulk and at interfaces, while ascribing fatigue to defect generation and domain-wall pinning²⁵.”

Lines 11-15, Page 13: “In a word, the reported rich phase transitions among *O*-phase and other non-polar phases under electron beam irradiation confirm the low energy barriers among these phases as reported previously³⁵, and further reveal the asynchronous sub-lattice distortion model governing these pathways. This model integrates the diverse and complex structural phase transition behaviors in HfO₂-based materials into a unified theoretical framework, systematically elucidating numerous controversies in prior researches and providing a deeper, atomistic understanding of phase transition mechanisms.”

Question and comment (R3.2): The observation of reversible FE–AFE transitions and T-O phase reversibility is highly relevant to device reliability. It would be helpful to quantify or discuss the kinetics and energy barriers of the observed transitions, at least qualitatively. (1) How do the observed phase transition timescales under e-beam irradiation compare with those under electric field cycling in actual devices? (2) Are these pathways accessible under field-induced switching, or are they artifacts of the beam-matter interaction? (3) Could the authors comment on how closely the electron-beam-induced pathways mimic the actual operational phase transitions in memory applications?

Reply to Question and comment (R3.2):

We thank the reviewer for these insightful questions regarding the relevance of our observed phase transitions to real device operation. We fully agree with the referee that the discussions about the kinetics and energy barriers of the observed transitions are crucial and helpful for our Manuscript.

As noticed by the referee, we have discussed multiple reversible phase transitions among the different polymorphs in HfO₂-based superlattice films, including the *O*-*M*₀₀₁, *O*-*M*₀₁₀, *O*-*T* and *FE*-*AFE* transitions.

Especially, the *O*-*M*₀₀₁ and *O*-*M*₀₁₀ transformation processes can be uniformly categorized as the structural transitions between the *O*-phase and *M*-phase. In Fig. 2, the initial structure of the superlattice thin film presents the nonpolar cubic lattice, which may correspond to either the *C*-phase or *T*-phase. Since the theoretical energy of the *C*-phase is much higher than that of *T*-phase^{12,20}, the initial structure of the film is considered to be *T*-phase. It is revealed that under electron beam irradiation, the *T*-phase tends to first transform into the *O*-phase, and then further into the *M*-phase. This transition pathway is consistent with previous reports of theoretical calculations. It was

revealed that, under fixed lattice parameters, the energy required for the T - O transformation was much lower than that for a direct transition from T -phase to M -phase²¹.

Regarding the phase transition between the T -phase and O -phase as shown in Fig. 3 in the Manuscript, it has been found that the T -phase preferentially transforms into the FE - O phase. In the reverse transformation, the AFE - O phase first transforms into the FE - O phase before transitioning to the T -phase. This behavior is attributed to the fact that the energy barrier for the transition between T -phase and FE - O phase is lower than that for the transition between T -phase and AFE - O phase^{15,21}.

With respect to the structural transition between different variants of the O -phase, namely FE - O and AFE - O , it is observed that under the same electron beam irradiation duration, the area of the FE - AFE transition is significantly larger than that of the AFE - FE transition (Fig. 4 in the Manuscript), indicating that the AFE - O phase occupies a lower energy state. This finding is consistent with previous theoretical calculations¹¹.

As a result, the multiple phase transitions are observed in our experiments.

(1) Comparison of timescales between e-beam irradiation and electric-field cycling:

The phase transition timescales observed under electron-beam irradiation (minutes to hours) are not directly comparable to those under electric-field cycling in functional devices (nanoseconds to sub-nanoseconds)²². This is because the driving forces are fundamentally different.

Electric-field switching is a fast, collective process driven by the coherent reorientation of dipoles under a strong, externally applied bias. However, electron-beam irradiation induces phase transitions primarily through localized electrostatic charging effect, which is a slower, more gradual process that allows us to stably image the intermediate states.

However, we emphasize that the atomistic pathways of phase transitions revealed by our electron-beam experiments are expected to be fundamentally similar to those under electric fields, as both are governed by the intrinsic energy landscape and crystal symmetry of the HfO_2 -based system^{11,15,21}. The value of our approach lies in using the slower electron-beam timescale to resolve these microscopic mechanisms that are otherwise temporally inaccessible in real-time device operation.

(2) Accessibility of pathways under electric field vs. beam-matter artifacts:

We believe the phase transition pathways we observed are not artifacts of beam-matter interaction, but rather intrinsic structural mechanisms that should also be

accessible under electric fields. This is supported by the following considerations. Firstly, the reversible phase transition between *FE-O* phase and *AFE-O* phase we observed is consistent with prior ex-situ TEM studies under electric fields¹³. Besides, the transition from *T*-phase to *O*-phase is also confirmed by previous DFT calculation results¹⁵. In addition, the studies of different cycling stages in the Gf:HfO₂ capacitors revealed that the wake-up stage was attributed to the phase changes, including the transformation from *M*-phase to *O*-phase in the bulk and the decrease of *T*-phase at the interface region¹⁹. Secondly, the polarization-direction-dependent transition pathways (e.g., $O_{\parallel} \rightarrow M$ vs. $O_{\perp} \rightarrow T$) reflect the intrinsic anisotropy of the *O*-phase, which should also influence field-induced switching. Finally, the transition pathways of *FE-AFE* and *T-O* transitions under e-beam we observed suggest the low energy barriers, consistent with the small energy differences predicted by DFT^{15,21}.

We acknowledge that shear-strain-assisted *O-M₀₁₀* transitions may be more readily triggered by local defects or strain under electron-beam irradiation. However, similar localized strain states can also be present in polycrystalline device films, induced by grain boundaries, electrode constraints, or cycling fatigue.

(3) Relevance to operational phase transitions in memory devices:

We believe the electron-beam-induced pathways closely mimic the essential phase transitions critical for memory device operation and reliability. On the one hand, the reversible *FE-AFE* transition is directly analogous to the polarization reversal and the wake-up/fatigue processes observed in HfO₂-based ferroelectric capacitors¹³. On the other hand, the transformations between the ferroelectric *O*-phase and the non-polar *T*- or *M*-phases are fundamental to understanding endurance degradation and phase instability, as the metastable *O*-phase can irreversibly decay into these non-polar phases during electric field cycling.

In conclusion, while the external stimulus (electron-beam vs. electric-field) differs, the structural transformation mechanisms we uncovered provide a crucial atomistic-level foundation for interpreting device-level phenomena. This understanding is vital for designing strategies to stabilize the ferroelectric phase and enhance the endurance of hafnia-based memory devices.

Changes in the revised Manuscript:

As suggested by the referee, we have added some discussions in the revised Manuscript.

Lines 15-19, Page 6: “The dominant structure in HZO layers in the as-grown

superlattice film (Fig. 2a) presents the nonpolar cubic lattice, which may correspond to either the *C*-phase or *T*-phase. Since the theoretical energy of the *C*-phase is much higher than that of *T*-phase³², the initial structure of the film is considered to be *T*-phase, which is also consistent with the XRD result in Fig. 1g. Figure 2a-f tracks the phase evolution among *T*-, *O*_{//}-, and *M*-phases in the HZO layer under electron beam irradiation.”

Lines 24-30, Page 6 and Lines 1-3, Page 7: “The predominant pathway is that *T*-phase progressively transforms into *O*_{//}-phase (pink mask) with increasing irradiation time, followed by the emergence of an intermediate state *M*₀₀₁' (green mask), and ultimately the formation of the *M*-phase projected along [001] direction (*M*₀₀₁, cyan mask). This transition pathway is consistent with previous reports of density functional theory calculations. It was revealed that the energy required for the *T*-*O* transformation was much lower than that for a direct transition from *T*-phase to *M*-phase under the fixed lattice parameters of HfO₂³³. Crucially, the transformation from *O*-phase to *M*₀₀₁-phase proceeds through an intermediary phase, with the *O*-phase first evolving into the *M*₀₀₁' phase before converting to the final *M*₀₀₁ phase. The *M*₀₀₁' phase is distinct from *M*₀₀₁ in its lattice distortion, as evidenced in Supplementary Fig. 10. Understanding the sequence of cationic (Hf/Zr) and anionic (O) sublattice distortions during this process is key to elucidating the underlying mechanism.”

Line 10-12, Page 11: “When transforming from the *T*-phase to the *O*_⊥-phase, the *T*-phase preferentially transforms into the *FE*-*O*_⊥ phase rather than *AFE*-*O*_⊥ phase (yellow rectangle in Supplementary Fig. 11b), which is supported by previous theoretical predictions that energy barrier for the transition between *T*-phase and *FE*-*O* phase is lower than that for the transition between *T*-phase and *AFE*-*O* phase^{33,34}.”

Lines 2-8, Page 12: “More critically, under identical irradiation durations, the phase-transition volume is larger for the *FE*-to-*AFE* transition (Figs. 4e-f) than for the reverse *AFE*-to-*FE* process (Figs. 4g-h). This asymmetry provides direct experimental evidence for the lower energy state and greater stability of the *AFE*-*O* phase, thereby validating earlier theoretical predictions^{14,34}. Furthermore, Fig. 1h shows that the *O*-phase is laterally confined within the HZO layers, whereas the adjacent ZO layers maintain the *T*-phase structure.”

Lines 29-30, Page 13 and Lines 1-4, Page 14: “Thirdly, the observed *FE*-*AFE* reversibility and *O*-phase instability towards non-polar phases provide a microscopic

basis for understanding device phenomena such as wake-up, fatigue, and endurance degradation. Therefore, these insights offer critical guidance for designing more reliable hafnia-based ferroelectric memories.”

Lines 11-20, Page 14: “It should be noted that this study focuses specifically on the mechanism of electron beam irradiation-induced structural phase transitions. While the timescales of electron beam irradiation-induced transitions are not directly comparable to the nanosecond-scale field-induced switching in devices, the atomistic pathways revealed here are expected to be fundamental. The asynchronous sublattice distortion mechanisms are governed by the intrinsic energy landscape of HfO₂-based materials, which also dictates the response under an electric field. Even though, we fully recognize that other types of stimuli may lead to differences in phase transition types and kinetic processes. Further research on this basis, particularly exploring the influence of *in-situ* electric fields under different application modes, is of significant importance.”

Question and comment (R3.3): The XRD data shown in Fig. 1g appears to be limited to a narrow angular range near the (002) reflection. For a more comprehensive understanding of the superlattice structure and to verify the presence of (001) reflection and other reflections related to the superlattice, I suggest that the authors provide an extended scan that includes the (001) region as well. If available, please also provide x-ray reflectivity. This would help confirm whether any superlattice-related periodicity or structural modulation is detectable in reciprocal space.

Reply to Question and comment (R3.3):

We appreciate the constructive suggestions from the referee. As noticed by the referee, we have provided the high-resolution XRD 2θ - ω scan of the superlattice film in Fig. 1g in the Manuscript. As suggested by the referee, we have provided the XRD scan result with wider range from 10°-80°, as shown in Fig. R3.1. The XRD pattern of the YSZ (001) substrate exhibits only the (002) and (004) diffraction peaks, with no presence of the (001) diffraction peak. This observation originates from the systematic extinction rules dictated by the specific space group (*Fm-3m*) of the YSZ crystal structure. In this space group, diffraction from (00L) planes requires the index L to be even. Since (001) and other odd-index planes such as (003) do not satisfy this condition, their diffraction peaks are entirely forbidden.

Fig. R3.1| XRD θ - 2θ scan from 10° - 80° for the superlattice film.

Furthermore, as suggested by the referee, we have also provided the ω - 2θ scan for the superlattice film to reveal the X-ray reflectivity (XRR), as shown in Fig. R3.2. The film thickness is determined to be approximately 17.6 nm by analyzing the XRR curve and calculating the oscillation period ($\Delta\theta = 0.23^\circ$) of the interference fringes. This result is in good agreement with the value of 19 nm obtained from atomic-scale HAADF-STEM measurements.

Fig. R3.2| XRD ω -2 θ scan for the superlattice film.

Changes in the revised Manuscript and revised Supplementary Materials:

We have added Fig. R3.1-R3.2 as Supplementary Fig. 5-6 in the revised Supplementary Materials. In the meanwhile, we have added some descriptions in the revised Manuscript.

Lines 6-11, Page 5: “High-resolution X-ray diffraction (XRD) scan (Fig. 1g) and wide-angle XRD scan (Supplementary Fig. 5) reveal the single-crystalline nature of the superlattice and also identified coexisting *M*-, *O*-, and *T*-phases within the film. Besides, the film thickness is calculated as 17.6 nm based on the X-ray reflectivity curve in Supplementary Fig. 6, which is in good agreement with the value of 19 nm obtained from atomic-scale HAADF-STEM measurements.”

Question and comment (R3.4): Experimental details

- a. If available, please provide the P-E hysteresis curves of pristine HZO samples in comparison with those of the irradiated counterparts to better illustrate the effect of electron beam exposure on ferroelectric switching.
- b. Provide experimental evidence or details regarding how the cation/anion stoichiometry was characterised in the HZO superlattice samples.

Reply to Question and comment (R3.4):

We appreciate the constructive suggestions from the referee. We fully agree with the referee that it is definitely crucial to illustrate the effect of electron beam exposure on ferroelectric switching by comparing the P-E hysteresis curves of pristine HZO samples and the irradiated counterparts. However, obtaining reliable P-E hysteresis curves in this study presents inherent difficulties due to the unique superlattice film structure of our samples, which can be primarily attributed to the following two reasons. On the one hand, the sample fabrication process did not involve the deposition of a bottom electrode, thereby precluding the possibility of standard capacitor-based ferroelectric testing. Although the interdigitated electrodes were used as an alternative approach, the in-plane electric field configuration is inherently more susceptible to leakage currents. On the other hand, the superlattice's extremely small thickness of approximately 19 nm resulted in significant leakage currents during electrical testing. This severely degraded the signal-to-noise ratio, completely overwhelming the weak ferroelectric polarization switching signal and thus making it impossible to acquire

comparable P-E loops for quantitative analysis. Such significant leakage in ultra-thin films without a bottom electrode is a recognized challenge within the field.

We sincerely apologize for being unable to provide the P-E hysteresis curves and plan to address this question in future work by designing high-quality hafnia-based superlattice samples with a capacitor structure to more directly investigate the influence of electron beam irradiation on their ferroelectric switching.

As noticed by the referee, we have provided the atomic-resolved EDS elemental map in Fig. 1c-e in the Manuscript for the superlattice film, with the EDS elemental map of Hf and Zr combined distribution and O distribution also being shown in Fig. R3.3a-b. Besides, the elemental spacing profile of Hf, Zr, and O corresponding to the white rectangle in Fig. R3.3a is also shown in Fig. R3.3c. In the ZrO_2 and $Hf_{0.5}Zr_{0.5}O_2$ layers, the distributions of Hf, Zr, and O elements are generally uniform and stable. However, a transition region exists between the two layers, exhibiting a gradual variation in elemental composition.

Fig. R3.3| Elemental distribution in the superlattice film. a, Atomic-resolved EDS elemental map of Hf and Zr combined distribution. **b,** Atomic-resolved EDS elemental map of O distribution. **c,** Elemental spacing profiles of Hf, Zr, and O corresponding to the white rectangle in (a).

Based on the results of the elemental distribution, the atomic ratios of the elements in ZrO_2 and $Hf_{0.5}Zr_{0.5}O_2$ can be determined, as shown in Tables R3.1 and R3.2. In the ZrO_2 layer (Table R3.1), the stoichiometric ratio of Zr to O is 27.8:65.4. Notably, about 6.8% Hf was detected in this layer, which is attributed to measurement errors or possible diffusion of Hf from the adjacent $Hf_{0.5}Zr_{0.5}O_2$ layers. In the $Hf_{0.5}Zr_{0.5}O_2$ layer (Table R3.2), the atomic ratio of Hf, Zr, and O is 15.4:18.9:65.7. With reference to the atomic ratio of Zr, a relative decrease in the atomic ratio of Hf was observed in the $Hf_{0.5}Zr_{0.5}O_2$

layer, which is expected to attribute to the diffusion of Hf into the adjacent ZrO₂ layer shown in Tables R3.1. Concurrently, a slight reduction in the atomic ratio of O was also displayed, a trend also reflected in the O distribution profile in Fig. R3.3c.

Table R3.1| The quantitative EDS results of O and Zr in the ZrO₂ layer of the superlattice film.

Element	Atomic fraction in ZrO ₂	Atomic error in ZrO ₂
O	65.4%	2.4%
Zr	27.8%	2.6%
Hf	6.8%	0.7%

Table R3.2| The quantitative EDS results of O, Zr and Hf in the Hf_{0.5}Zr_{0.5}O₂ layer of the superlattice film.

Element	Atomic fraction in Hf _{0.5} Zr _{0.5} O ₂	Atomic error in Hf _{0.5} Zr _{0.5} O ₂
O	65.7%	2.0%
Zr	18.9%	2.0%
Hf	15.4%	1.4%

Changes in the revised Manuscript and revised Supplementary Materials:

We have added Fig. R3.3, Table R3.1 and Table R3.2 as Supplementary Fig. 1, Supplementary Tab. 1 and Supplementary Tab. 2 in the revised Supplementary Materials. In addition, we have added some discussions in Lines 25-30, Page 4 in the revised Manuscript.

Lines 25-30, Page 4: “Atomic-scale EDS-STEM mapping revealed sharp interlayer interfaces (Fig. 1c-e), attesting to the high-quality growth. **The in-depth analysis of the cation/anion stoichiometry for the superlattice film is further shown in Supplementary Fig. 1 and Supplementary Tab. 1-2. The results indicate a slight interdiffusion of Hf into the neighboring ZrO₂ layer, along with a minor oxygen deficiency in the HZO layer. This slight off-stoichiometry and elemental interdiffusion are likely a consequence of the high-temperature growth process employed for the**

superlattice film.”

Furthermore, we have added some descriptions about the details of quantitative stoichiometric analysis in the Methods part.

“TEM sample preparation, TEM observation. The cross-sectional TEM samples for STEM observation were prepared by traditional process: slicing, gluing, grinding, dimpling and finally ion milling. A Gatan 691 PIPS was used for ion milling. During the ion milling process, a low angle (5°) and a cooling stage were used firstly, and the final ion milling voltage was 0.3 eV for 10 minutes to reduce the beam damage. The selected area electron diffraction images were recorded using a conventional TEM (JEOL JEM-F200 working at 200 kV). HAADF-STEM, iDPC-STEM and EDS-STEM images were recorded using Spectra 300 X-FEG aberration-corrected scanning transmission electron microscope (ThermoFisher Scientific) with double aberration (Cs) correctors and a monochromator operating at 300 kV. For the acquisition of HAADF-STEM and iDPC-STEM images, a spotsize of 9 and a beam current of 100 pA were used; for EDS-STEM mapping, a spotsize of 6 and a beam current of 100 pA were employed.”

“STEM result analyses. Atom positions were accurately determined using 2D Gaussian peak fitting in Matlab³⁶, thus making it possible to acquire the information of the Hf/Zr- and O- atomic positions, Hf/Zr-ionic in-plane and out-of-plane rotation and O-ionic displacement. It should be noted that a wiener filter of HAADF and iDPC-STEM and a low-pass annular mask restricted to the instrument resolution limit of the images were used to reduce the noise of the obtained images. Quantitative stoichiometric analysis was conducted in Velox 3.14.0 using atomic-resolved EDS-STEM data, by applying the Empirical model for background correction and the Brown-Powell model for the ionization cross-section.”

Reference

1. Lee, H.-J. et al. Scale-free ferroelectricity induced by flat phonon bands in HfO₂. *Science* **369**, 1343-1347 (2020).
2. Ramesh, R., Schlom, D. G. Creating emergent phenomena in oxide superlattices.

- Nature Reviews Materials* **4**, 257-268 (2019).
3. Ho Nyung, L., Christen, H. M., Chisholm, M. F., Rouleau, C. M., Lowndes, D. H. Strong polarization enhancement in asymmetric three-component ferroelectric superlattices. *Nature* **433**, 395-399 (2005).
 4. Bousquet, E. et al. Improper ferroelectricity in perovskite oxide artificial superlattices. *Nature* **452**, 732-736 (2008).
 5. Ahluwalia, R. et al. Manipulating ferroelectric domains in nanostructures under electron beams. *Physical Review Letters* **111**, 165702 (2013).
 6. Chen, Z., Wang, X., Ringer, S. P., Liao, X. Manipulation of nanoscale domain switching using an electron beam with omnidirectional electric field distribution. *Physical Review Letters* **117**, 027601 (2016).
 7. Ma, J. Y. et al. Real-time observation of phase coexistence and a_1/a_2 to flux-closure domain transformation in ferroelectric films. *Acta Materialia* **193**, 311-317 (2020).
 8. Wei, J. et al. Direct imaging of atomistic grain boundary migration. *Nature Materials* **20**, 951-955 (2021).
 9. Egerton, R. F., Li, P., Malac, M. Radiation damage in the TEM and SEM. *Micron* **35**, 399-409 (2004).
 10. Jiang, N. Electron beam damage in oxides: A review. *Reports on Progress in Physics* **79**, 016501 (2016).
 11. Li, X. et al. Ferroelastically protected reversible orthorhombic to monoclinic-like phase transition in ZrO_2 nanocrystals. *Nature Materials* **23**, 1077-1084 (2024).
 12. Li, X. et al. Polarization switching and correlated phase transitions in fluorite-structure ZrO_2 nanocrystals. *Advanced Materials* **35**, 2207736 (2023).
 13. Cheng, Y. et al. Reversible transition between the polar and antipolar phases and its implications for wake-up and fatigue in HfO_2 -based ferroelectric thin film. *Nature Communications* **13**, 1-8 (2022).
 14. Guan, S.-H., Zhang, X.-J., Liu, Z.-P. Energy landscape of zirconia phase transitions. *Journal of the American Chemical Society* **137**, 8010-8013 (2015).
 15. Liu, S., Hanrahan, B. M. Effects of growth orientations and epitaxial strains on phase stability of HfO_2 thin films. *Physical Review Materials* **3**, 054404 (2019).
 16. Zhou, S., Zhang, J., Rappe, A. M. Strain-induced antipolar phase in hafnia stabilizes robust thin-film ferroelectricity. *Science Advances* **8**, eadd5953 (2022).
 17. Wang, S. et al. Unlocking the phase evolution of the hidden non-polar to ferroelectric transition in HfO_2 -based bulk crystals. *Nature Communications* **16**, 3745 (2025).
 18. Pešić, M. et al. Physical mechanisms behind the field-cycling behavior of HfO_2 -based ferroelectric capacitors. *Advanced Functional Materials* **26**, 4601-4612 (2016).
 19. Grimley, E. D. et al. Structural changes underlying field-cycling phenomena in ferroelectric HfO_2 thin films. *Advanced Electronic Materials* **2**, 7 (2016).
 20. Reyes-Lillo, S. E., Garrity, K. F., Rabe, K. M. Antiferroelectricity in thin-film ZrO_2 from first principles. *Physical Review B* **90**, 140103(R) (2014).
 21. Xu, X. et al. Kinetically stabilized ferroelectricity in bulk single-crystalline $HfO_2:Y$. *Nature Materials* **20**, 826-832 (2021).
 22. Zhou, C. et al. Enhanced polarization switching characteristics of HfO_2 ultrathin

films via acceptor-donor co-doping. *Nature Communications* **15**, 2893 (2024).

Reply to referees' questions and comments:

Ref number: NCOMMS-25-68598A

Title: Roadmap of phase transitions in hafnia-based superlattice films

Authors: Wan-Rong Geng, Bo-Rui Wang, Yin-Lian Zhu, Si-Rui Zhang, Min Liao, Xiu-Liang Ma

24 February, 2026

Reply to Reviewer #2 (R2):

Remark: The authors have responded to all comments thoroughly and revised the manuscript accordingly. Therefore, I recommend this manuscript for publication.

Reply to the remark:

We sincerely thank the reviewer for their positive feedback and for recommending our manuscript for publication.

Reply to Reviewer #3:

We appreciate the positive comment by the referee that “The referee recognizes the authors’ efforts in addressing most of the comments and appreciates the expanded discussion of the asynchronous sublattice distortion model”, that “The additional reference, revised figures, and more detailed explanations improve the clarity and depth of the manuscript”, and that “The manuscript addresses a significant and timely topic, and the concept of asynchronous sublattice distortion is intriguing”.

In the meanwhile, the referee also raises several specific questions and comments which are summarized into four aspects. We fully understand the referee’s concerns, and here we have addressed all the questions and discussed all the comments one by one in the following. The revisions are written in **RED** in the revised manuscript and the revised Supplementary Materials.

Question and comment (R3.1): The authors claim that the film is “a (Hf_{0.5}Zr_{0.5}O₂)₆ superlattice with a period of 7 unit cells per layer on a [001]-oriented YSZ substrate” in the revised manuscript, as opposed to 6 unit cells. Revised Supplementary Fig. S7, however, presents a different film with a 5 unit-cell periodicity, resulting in a discrepancy in the sample description. The referee recommends that this ambiguity be definitively addressed at the beginning of the manuscript.

Reply to Question and comment (R3.1):

We appreciate the kind reminder from the referee. We have confirmed that the thickness of one (Hf_{0.5}Zr_{0.5}O₂/ZrO₂) growth period in the superlattice film primarily discussed in this Manuscript corresponds to 6 unit cells based on the atomic-resolved HAADF-STEM image (Fig. 1b) and EDS-STEM mapping (Fig. 1c-e). We have updated relevant descriptions in the revised manuscript.

It is worthwhile to note that the additional superlattice films of (HZO-ZO-5)₆ and (HZO-ZO-11)₆ are provided to compare with the (HZO-ZO-6)₆ superlattice film mainly discussed in the Manuscript. As revealed by Fig. 1h and Fig. 2 in the Manuscript, the (HZO-ZO-6)₆ superlattice film exhibits a coexistence of orthorhombic (*O*), tetragonal (*T*), and monoclinic (*M*)-phases, with the *O*-phase featuring both in-plane (*O*_{//}) and out-of-plane (*O*_⊥) polarization orientations. In contrast, the (HZO-ZO-5)₆ film displays the uniform structure of *T*-phase without the existence of *O*-phase (Supplementary Fig. 8). However, despite sharing the same character of multiphase coexistence as (HZO-ZO-6)₆ film, the (HZO-ZO-11)₆ superlattice film shows the layered phase structure (Supplementary Fig. 9). The first two HZO layers adopt the *M*-phase oriented along

different zone axes. The third layer exhibits the $O_{//}$ -phase and the fourth layer displays the coexistence of $O_{//}$ -phase and O_{\perp} -phase, then the fifth layer stabilizes the O_{\perp} -phase (Supplementary Fig. 9). As a result, the thickness of one (HZO-ZO) growth period plays the crucial role on determining the phase structure, spatial distribution, and epitaxial quality of the (HZO-ZO)₆ superlattice films. Actually, in the manuscript, we had discussed the influences of growth period thickness in Lines 1-3, Page 6.

Lines 1-3, Page 6: “It is revealed that the phase structure, spatial distribution, and epitaxial quality of the superlattice films depend on the thickness of one (HZO-ZO) period (Supplementary Fig. 8-9).”

To clarify the sample information involved in the present work, we have added some descriptions in Lines 23-28, Page 4, corresponding to the beginning of the revised main text.

Lines 23-28, Page 4: “Thus, using pulsed laser deposition (PLD), we fabricated a series of high-quality (Hf_{0.5}Zr_{0.5}O₂-ZrO₂)₆ ((HZO-ZO)₆) epitaxial superlattice thin films on [001]-oriented YSZ substrates with different thickness of one (HZO-ZO) growth period, including the 5 unit cells (labelled as HZO-ZO-5), 6 unit cells (labelled as HZO-ZO-6) and 11 unit cells (labelled as HZO-ZO-11). In Fig. 1a-1b, the (HZO-ZO-6)₆ superlattice film is displayed.”

Furthermore, we have added some sentences in the Method part.

“Using pulsed laser deposition with a Coherent ComPex PRO 201 F KrF ($\lambda = 248$ nm) excimer laser, a series of (Hf_{0.5}Zr_{0.5}O₂-ZrO₂)₆ superlattice films were deposited on YSZ (001) substrates, including the (Hf_{0.5}Zr_{0.5}O₂-ZrO₂-5)₆, (Hf_{0.5}Zr_{0.5}O₂-ZrO₂-6)₆, and (Hf_{0.5}Zr_{0.5}O₂-ZrO₂-11)₆.”

Question and comment (R3.2): Furthermore, the XRD and XRR data currently shown are not sufficient to verify superlattice periodicity and structural modulation. While the authors provide XRR data above 0.6° , the signal only shows a single broad oscillation, without distinct superlattice reflection peaks. In comparison, previous studies [refer to Li et al., Nat. Comm. 16, 6417 (2025)] demonstrate well-resolved superlattice peaks in XRR.

The referee recommends that the authors conduct synchrotron-based XRD and XRR studies to resolve the superlattice peaks and enhance the signal-to-noise ratio. This would help whether there are detectable structural modulations or symmetry changes after e-beam irradiation.

This data is essential to examine if structural modulation in the forbidden (001) reflection region, where substrate signal is suppressed, indicates any modifications

before and after electron beam irradiation, which is pivotal to the manuscript's claims. Given that the authors report clear asymmetric sublattice distortions and complex phase transitions at the atomic scale via STEM, it is crucial to figure out if these structural modifications also lead to measurable signals in reciprocal space. If such distortions are intrinsic and widespread inside the irradiated volume, they should appear as modulations in the XRD pattern, particularly in areas of reciprocal space that are often background-suppressed. See also Solomon et al., Sci. Adv. 11, eadq5943 (2025) for an example where beam-induced structural changes in zirconia were clearly evident in XRD analyses.

Reply to Question and comment (R3.2):

We appreciate the constructive suggestions from the referee. Although we currently do not have access to synchrotron facilities to perform the suggested measurements, we have carefully optimized the laboratory XRD/XRR experimental conditions to improve intensity and reduce the background signal. As a result, we have obtained high-resolution θ - 2θ XRD scan (Fig. R3.1) and X-ray reflectivity (XRR) measurement (Fig. R3.2) of the superlattice film with significantly enhanced signal-to-noise ratio, which allow a more reliable analysis of the superlattice periodicity and film thickness. Based on these results, we have gained a deeper understanding of the superlattice films.

As shown in Fig. R3.1, The 0th-order superlattice reflection (SL0), -1 and +1 satellite peaks (SL-1 and SL+1) are highlighted, at $2\theta \approx 35.2^\circ$, 32.5° and 37.9° , respectively. The satellite peak separation is consistent with the designed superlattice growth periodicity of 3.3 nm, confirming the formation of a well-defined chemically modulated multilayer structure. Furthermore, a series of weak Laue oscillations denoted by “*” arise from the coherent interfaces and finite film thickness, indicating a smooth and uniform superlattice film. Although the superlattice is known to host mixed monoclinic (*M*-), orthorhombic (*O*-), and tetragonal (*T*-) phases based on the atomic-resolved STEM results, distinct diffraction peaks associated with individual phases are not clearly resolved, especially the *O*-phase and *T*-phase. This is mainly attributed to that the strong substrate peak together with superlattice interference effects (satellites peaks and Laue oscillations) suppresses or overlaps with the weak phase-specific contributions. Therefore, the XRD pattern is dominated by the superlattice periodicity and thickness-related interference features, rather than separate phase-resolved reflections. Thus, we have updated the XRD result in Fig. 1g with Fig. R3.1 and also the descriptions in the revised Manuscript.

Fig. R3.1 High-resolution symmetric XRD θ - 2θ scan ($2\theta = 30$ - 42°) of the $(\text{HZO})_6$ superlattice film grown on a YSZ (001) substrate.

Besides, as shown in the XRR pattern (Fig. R3.2), clear Kiessig fringes indicate the high-quality growth of the superlattice film. From the period of the Kiessig fringes ($\Delta 2\theta \approx 0.43^\circ$), the total film thickness is estimated to be approximately 20.5 nm. The distinct superlattice Bragg peak observed at $2\theta = 2.67^\circ$ corresponds to a superlattice period of about 3.3 nm, indicating a well-defined periodic chemical modulation. The ratio between the total thickness and the superlattice period suggests approximately six superlattice repeats, consistent with the designed multilayer architecture.

Fig. R3.2 The XRR pattern of the superlattice film.

We thank the reviewer for suggesting the synchrotron XRD study and for pointing out the related work by Solomon et al. (Sci. Adv. 11, eadq5943 (2025)), where the phase evolution of ion-irradiated ZrO_2 nanocrystals was clearly resolved by synchrotron XRD. However, it should be emphasized that their study focuses on a nanocrystalline system, which is fundamentally different from the epitaxial $(HZO-ZO)_6$ superlattice films grown on YSZ (001) investigated in our work. For epitaxial superlattice thin films, the symmetric XRD signal is typically dominated by pronounced superlattice satellite peaks and Laue oscillations, while diffraction signatures associated with individual polymorphs (monoclinic/orthorhombic/tetragonal) are often significantly suppressed or obscured. Therefore, XRD measurements are not sufficient to extract the complete structural evolution of the superlattice film before and after the electron-beam irradiation.

Moreover, in our Manuscript, it is demonstrated that different initial structural states can undergo distinct phase-transition pathways under electron-beam irradiation (see Fig. 5 in the Manuscript). As a consequence, even if the weak phase-related diffraction signals could be detected, they reflect the averaged response over multiple phase transformation behaviors within the film, making it difficult to resolve the differences among individual local phase-transition processes.

As a result, to clearly reveal the irradiation-induced structural evolution in reciprocal space, we performed FFT analyses of the atomic-resolved iDPC-STEM images acquired before and after the corresponding phase-transition processes. This approach enables us to capture characteristic diffraction features associated with local structural evolution, thereby providing a more reliable and spatially resolved description of the structural changes corresponding to different transformation pathways.

For the phase-transition processes among the T -phase, O -phase, and M_{001} -phase revealed in Fig. 2a-2i of the main text, we performed the FFT analyses corresponding to the atomic-resolution iDPC-STEM images of both the initial state and after 240 min of electron-beam irradiation. As shown in Figs. R3.3c and R3.3e, which correspond to the left regions in Figs. R3.3a and R3.3b, respectively, the structure evolves from the initial O -phase to a coexistence state of the O -phase and the M_{001}' -phase. Since the M_{001}' -phase exhibits structural characteristics similar to those of the O -phase, this transition does not lead to an obvious difference in reciprocal space. Thus, the FFT patterns in Figs. R3.3c and R3.3e remain essentially unchanged. In contrast, for the structural evolution in the right regions of Fig. R3.3a-b, the film structure transforms from the initial T -phase to the M_{001} -phase, which is clearly reflected in the corresponding FFT patterns. As shown in Figs. R3.3d and R3.3f, new diffraction spots emerge at the positions marked by the white circles, while the intensities of the diffraction spots marked by the orange circles exhibit noticeable changes. These features indicate a pronounced evolution in reciprocal space, consistent with the structural phase transition observed in real space.

For the transition between the O -phase and the M_{010} -phase shown in Fig. 2j-2o of the main text, the corresponding structural change can also be clearly identified in the FFT patterns. As shown in Fig. R3.4d, the characteristic diffraction spots highlighted by the orange parallelogram provide clear evidence of the reciprocal-space difference before and after the phase transition.

In addition, Fig. R3.5 shows the reversible transformation between the T -phase and the O -phase under electron-beam irradiation, corresponding to the structural evolution presented in Fig. 3 of the main text. It should be noted that throughout the irradiation process, the regions shown in Fig. R3.5a-d remain in a coexistence state of the T -phase and O -phase, with the primary difference being the variation in their relative fractions. Therefore, no obvious emergence or disappearance of diffraction spots is observed in the corresponding FFT patterns, and the overall reciprocal-space variation is relatively weak.

Furthermore, Fig. R3.6 presents the structural phase transition between the $FE-O$ phase and the $AFE-O$ phase under electron-beam irradiation, corresponding to the structural evolution shown in Fig. 4e-f of the main text. Since the $FE-O$ and $AFE-O$ phases possess different modulation periodicities, the FFT pattern of the $AFE-O$ phase exhibits additional modulation spots compared with that of the $FE-O$ phase, as indicated by the orange arrows in Fig. R3.6d. This provides clear reciprocal-space evidence for the structural difference between the two phases.

Therefore, by analyzing the FFT patterns, the reciprocal-space signatures associated with different phase-transition behaviors are effectively revealed.

Fig. R3.3 Phase transition among T -phase, O -phase and M_{001} -phase. a-b, Atomic-resolved iDPC-STEM images before and after electron beam irradiation, corresponding to Figs. 2a and 2f, respectively. c-d, Fast Fourier transformation (FFT) patterns for the orange and cyan rectangles of (a). e-f, FFT patterns for the orange and cyan rectangles of (b).

Fig. R3.4 Phase transition between *O*-phase and *M₀₁₀*-phase. **a-b**, Atomic-resolved iDPC-STEM images before and after electron beam irradiation, corresponding to Figs. 2j and 2l, respectively. **c-d**, FFT patterns for the orange rectangles in (a) and (b), respectively.

Fig. R3.5 Phase transition between *T*-phase and *O*-phase. **a-b**, Atomic-resolved iDPC-STEM images after different irradiation durations: 0 min (a); 5 min (b); 15 min (c); 60 min (d), corresponding to Figs. 3a, 3b, 3f, and 3g, respectively. **e-h**, Corresponding FFT patterns of (a-d).

Fig. R3.6 Phase transition between *AFE-O* phase and *FE-O* phase. **a-b**, Atomic-resolved iDPC-STEM images before and after electron beam irradiation, corresponding to Figs. 4e and 4f, respectively. **c-d**, Corresponding FFT patterns of (a) and (b), respectively.

Changes in the revised Manuscript and revised Supplementary Materials:

We have updated the Fig. 1g with Fig. R3.1 in the revised Manuscript and also updated the Supplementary Fig. 6 with Fig. R3.2 in the revised Supplementary

Materials. In addition, we have updated the corresponding description of Fig. 1g and Supplementary Fig. 6 in Lines 13-19, Page 5 in the revised Manuscript.

Lines 13-19, Page 5: High-resolution X-ray diffraction (XRD) scan (Fig. 1g) and wide-angle XRD scan (Supplementary Fig. 5) reveal the single-crystalline nature of the superlattice and the well-defined superlattice periodicity and high-quality film growth. Besides, the X-ray reflectivity in Supplementary Fig. 6 shows clear Kiessig fringes further confirming the high-quality growth of the superlattice film³⁰. From the period of the Kiessig fringes, the total film thickness is estimated to be approximately 20.5 nm. The distinct superlattice Bragg peak observed at $2\theta = 2.67^\circ$ corresponds to a superlattice period of about 3.3 nm, indicating a well-defined periodic chemical modulation.

In addition, we have added the recommended paper (Solomon et al., Sci. Adv. 11, eadq5943 (2025)) as Ref. 28 in the revised Manuscript. Another recommended paper (Li et al., Nat. Comm. 16, 6417 (2025)) had been cited as Ref. 30 in our Manuscript.

Question and comment (R3.3): The referee agrees with the authors' explanation that it was not possible to measure P-E hysteresis because there was a lot of leakage current in the superlattice structure with in-plane interdigitated electrodes. While the experimental challenge is understandable, this constraint is especially significant due to the manuscript's focus on structural phase transitions closely linked to the presence of polar or ferroelectric phases.

In HfO₂-based systems, reliable P-E measurements have been shown even in ultrathin films, typically due to their large coercive fields. The inability to evaluate switching characteristics in the current samples raises concerns regarding material quality or structural defects, particularly given that the proposed phase transitions involve noticeable structural distortions and the emergence of potential defective states.

The referee recommends that the authors provide current density-voltage (J-V) or leakage current data measured in the same in-plane interdigitated configuration. Even in the absence of a measurable switching signal, such data would allow readers to independently assess the leakage behavior and evaluate its implications for the functional quality of the film, particularly given the structural distortions reported under electron beam irradiation.

Reply to Question and comment (R3.3):

We appreciate the constructive suggestion from the referee. We fully agree with the referee that the current density-voltage (J-V) or leakage current data is crucial to assess the leakage behavior and evaluate its implications for the functional quality of the film. As suggested by the referee, we have provided the leakage current curve as a

function of electric field (J-E) in Fig. R3.7. The leakage current density measured using in-plane interdigitated electrodes exhibits a highly symmetric J-E behavior with respect to the electric field polarity. Over most of the measured field range, the current density remains nearly field-independent at the level of $\sim 10^{-4}$ A/cm². The field-independent behavior of the current density is nearly consistent with previous report on (HfO₂)₃/(ZrO₂)₃ superlattice film¹.

Fig. R3.7 Leakage current curve as a function of electric field (J-E).

Changes in the revised Manuscript and revised Supplementary Materials:

We have added Fig. R3.7 as Supplementary Fig. 10 in the revised Supplementary Materials. In addition, we have updated the description in Lines 6-8, Page 6 in the revised Manuscript.

Lines 6-8, Page 6: This inherent phase complexity can lead to poor **ferroelectric** performance **with evident leakage current (Supplementary Fig. 10)** and pose a risk of field-induced phase transitions causing device failure.

Question and comment (R3.4): As a complementary approach, the authors can consider techniques such as second harmonic generation (SHG) to probe the presence of polar orders.

Reply to Question and comment (R3.4):

We appreciate the constructive suggestion from the referee. As suggested, we have performed the second harmonic generation (SHG) measurement to confirm the presence of polar orders in the superlattice film.

Fig. R3.8 SHG characterizations of the superlattice film. a, Power-dependent SHG spectra. **b,** The excitation power dependence of SHG intensity with the coefficient fitted to 2.04671. **c,** Azimuth-dependent SHG signals of the superlattice film along parallel and perpendicular directions.

A wavelength of 1080 nm was used as the excitation laser for the SHG measurements. As shown in Fig. R3.8a, the SHG signal from the superlattice film increases monotonically with increasing excitation power. The dependence of the SHG intensity on the excitation power is analyzed in Fig. R3.8b, where a power-law fitting yields a slope of 2.04671, which is close to the theoretical value of 2 for a second-order nonlinear optical process. This result confirms that the observed signal originates from SHG and indicates a non-centrosymmetric nonlinear optical response of the film.

Figure R3.8c shows the polarization-dependent SHG intensity of the superlattice film. A clear angular modulation with twofold rotational symmetry is observed in both the parallel and perpendicular detection configurations. The pronounced polarization dependence suggests an in-plane anisotropy of the effective second-order nonlinear susceptibility. Such anisotropic SHG behavior is consistent with a broken inversion symmetry in the effective optical response, which may be associated with a polar crystal structure and/or preferentially oriented polar domains in the film.

Changes in the revised Manuscript and revised Supplementary Materials:

We have added Fig. R3.8 as Supplementary Fig. 7 in the revised Supplementary Materials. In addition, we have one sentence in Lines 24-25, Page 5 in the revised Manuscript.

Lines 24-25, Page 5: Furthermore, the iDPC-STEM image reveals two dominant phase structures in the film: a *T*-phase and an *O*-phase. **The presence of the polar phase is further confirmed by the second harmonic generation results in Supplementary Fig. 7.**

Reference

1. Li, J. et al. Enhancing ferroelectric stability: wide-range of adaptive control in epitaxial HfO₂/ZrO₂ superlattices. *Nat. Commun.* **16**, 6417 (2025).